# HALLUCINATION BENCHMARK FOR SPEECH FOUNDATION MODELS

## ABSTRACT

Hallucinations in automatic speech recognition (ASR) systems refer to fluent and coherent transcriptions produced by neural ASR models that are completely unrelated to the underlying acoustic input (i.e., the speech signal). While similar to conventional decoding errors in potentially compromising the usability of transcriptions for downstream applications, hallucinations can be more detrimental due to their preservation of syntactically and semantically plausible structure. This apparent coherence can mislead subsequent processing stages and introduce serious risks, particularly in critical domains such as healthcare and law. Conventional evaluation metrics are primarily centered on error-based metrics and fail to distinguish between phonetic inaccuracies and hallucinations. Consequently, there is a critical need for new evaluation frameworks that can effectively identify and assess models with a heightened propensity for generating hallucinated content. To this end, we introduce SHALLOW, the first benchmark framework that systematically categorizes and quantifies hallucination phenomena in ASR along four complementary axes: lexical, phonetic, morphological, and semantic. We define targeted metrics within each category to produce interpretable profiles of model behavior. Through evaluation across various architectures and speech domains, we have found that SHALLOW metrics correlate strongly with word error rate (WER) when recognition quality is high (i.e., low WER). Still, this correlation weakens substantially as WER increases. SHALLOW, therefore, captures fine-grained error patterns that WER fails to distinguish under degraded and challenging conditions. Our framework supports specific diagnosis of model weaknesses and provides feedback for model improvement beyond what aggregate error rates can offer. `anonymous.4open.science/r/SHALLOW`

## 1 INTRODUCTION

Automatic Speech Recognition (ASR) has made significant progress through large-scale foundation models, delivering high transcription accuracy across diverse domains Abouelenin et al. (2025a); Radford et al. (2023b); Puvvada et al. (2024); Rekesh et al. (2023). Despite these advancements, modern ASR systems still produce hallucinations: plausible content not grounded in the input audio Atwany et al. (2025); Frieske & Shi (2024).

Hallucinations create serious concerns in applications such as healthcare, legal transcription, and education, where transcription fidelity directly affects critical decisions Koenecke et al. (2024).

Hallucination phenomena have been widely studied across different AI domains, each with distinct characteristics and evaluation challenges. In text generation, hallucinations primarily concern factual inconsistencies where models produce statements that appear plausible but contradict verifiable truths Huang et al. (2025); Du et al. (2024). Similarly,

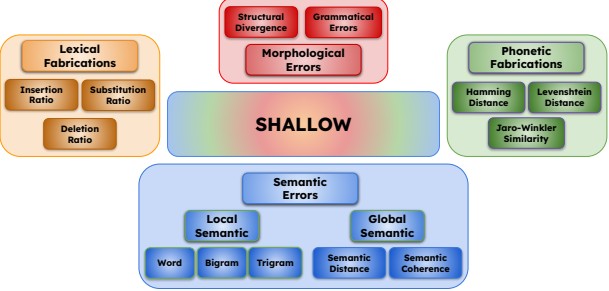

Figure 1: SHALLOW benchmark, with its four dimensions: lexical, phonetic, morphological, and semantic scores.

in text-to-image generation, hallucinations occur when models create visually coherent elements that weren't specified in the prompt or misrepresent requested objects Lim et al. (2025).

ASR hallucinations present a fundamentally different evaluation challenge: the key concern is fidelity to the acoustic signal, rather than factuality with respect to world knowledge or prompt adherence. This distinction is critical because ASR systems operate at the interface between signal processing and language modeling, where the ground truth is the spoken content rather than external knowledge. Unlike text or image generation, where external knowledge sources or prompt-image alignment can be used to verify factuality, ASR hallucinations require evaluation frameworks that specifically measure deviation from the actual spoken content. This unique characteristic necessitates specialized metrics beyond those used in other generative AI domains, particularly as ASR systems increasingly incorporate generative language capabilities that may prioritize fluency over acoustic fidelity Atwany et al. (2025); Kim et al. (2021b).

The standard evaluation metric for ASR systems, Word Error Rate (WER), provides a valuable aggregate measure of transcription accuracy. While effective for overall performance assessment, WER treats all errors as equivalent, without distinguishing between surface-level errors and critical semantic alterations Kim et al. (2021b). For example, transcribing "*take the medication*" as "*skip the medication*" results in just one word error according to WER, yet it completely reverses the meaning, with potentially harmful consequences in medical contexts.

Despite growing recognition of the hallucination problem in speech recognition Frieske & Shi (2024), the research community lacks standardized methods to systematically categorize and measure these phenomena. This gap limits both precise model assessment and targeted improvement efforts, as developers can't detect specific error types or assess how architectural changes impact them. Current evaluation practices that rely solely on aggregate metrics, e.g., WER, obscure meaningful differences in model behavior that significantly impact trustworthiness for specific applications.

To this end, we introduce SHALLOW (**S**peech **HALL**ucination **O**vervie**W**), a benchmark framework that decomposes ASR errors into four complementary dimensions (Figure 1): (1) *Lexical Fabrications* - content with no ground in the input audio, measured through insertion, substitution, and deletion ratios at a lexical level; (2) *Phonetic Fabrications* - errors where the model generates phonetically similar but lexically incorrect words, measured using metaphone-based distance metrics; (3) *Morphological Errors* - structural and grammatical distortions that alter the linguistic form of the transcription; and (4) *Semantic Errors* - meaning alterations captured at both local and global levels, measuring how semantic content is preserved or distorted.

The above-mentioned categories form the basis of the SHALLOW evaluation framework, which we apply across models and datasets. Hallucination behaviors are not uniformly distributed but rather reflect fundamental architectural design choices. Encoder-decoder models like Whisper (Large-v2/v3 Radford et al. (2023b)) demonstrate balanced error patterns across phonetic, morphological, and semantic dimensions, avoiding extreme trade-offs in any single category while maintaining strong overall accuracy. In contrast, decoder-based models (e.g., Phi-4-Multimodal-Instruct Abouelenin et al. (2025b)) incorporate stronger language modeling components that prioritize linguistic fluency over exact acoustic matching. This architectural difference leads them to achieve better performance in morphological and semantic dimensions while introducing more phonetically plausible substitutions (see Table 2). The distribution of scores across dimensions reveals the trade-off between acoustic fidelity and linguistic coherence, revealing differences that WER alone obscures. SHALLOW enables researchers to isolate specific hallucination categories affected by architectural or data changes, supporting targeted model development and alignment with application-specific requirements.

Statistical analysis across models and domains reveals that SHALLOW metrics correlate with WER under high-quality recognition conditions (i.e., low WER), but this relationship weakens significantly as transcription quality degrades. This breakdown highlights the capability of SHALLOW to capture fine-grained and type-specific hallucinations that WER alone cannot differentiate, particularly in acoustically challenging or out-of-distribution scenarios.

The key contributions of the present work include: (1) A structured taxonomy of ASR hallucination types grounded in linguistic and acoustic distinctions, with clear, quantifiable metrics for each category; (2) A standardized evaluation framework that enables comparison and diagnosis beyond aggregate accuracy scores; (3) An in-depth analysis demonstrating that SHALLOW metrics reveal error structure that WER alone cannot, particularly in acoustically challenging en-

Table 1: Examples of synthetic data, one per category type, with WER and SHALLOW metrics.

| Category | Description | Reference | Hypothesis | WER | LF | PF | ME | SE |
|---|---|---|---|---|---|---|---|---|
| **Lexical** | Adds unrelated or hallucinated words | They are playing chess outside | They are playing chess outside with magical stones | 0.60 | 0.19 | 0.31 | 0.15 | 0.29 |
| **Phonetic** | Substitutes with phonetically similar but incorrect words | I went to the retirement party | I bent to the retirement party | 0.17 | 0.05 | 0.04 | 0.27 | 0.13 |
| **Morphological** | Tense or agreement errors | They sing together every morning | They sings together every mornings | 0.40 | 0.12 | 0.02 | 0.40 | 0.08 |
| **(Local) Semantic** | Replaces a single word, changing the meaning | He painted the fence | He destroyed the fence | 0.25 | 0.08 | 0.37 | 0.34 | 0.66 |
| **(Global) Semantic** | Changes sentence meaning | They went to the beach for vacation | They stayed home for vacation | 0.57 | 0.14 | 0.45 | 0.36 | 0.68 |
| **Mixed Errors** | Combines lexical, morph. and semantic hallucinations | She fixed her broken glasses | She fix broken lens with dragon spark | 1.20 | 0.38 | 0.51 | 0.40 | 0.39 |
| **WER only** | High WER, same meaning | They joined us for dinner | They came over to eat with us | 1.20 | 0.38 | 0.64 | 0.40 | 0.17 |

vironments; (4) Evidence that SHALLOW enables targeted diagnosis of model behavior, supporting application-specific model selection and more informed iteration on ASR system design.

SHALLOW helps researchers diagnose specific model weaknesses, select models based on application-specific requirements, and measure targeted progress in reducing different types of errors. It represents an important step toward a more nuanced evaluation of ASR systems.

**Benchmark validation.** To validate the interpretability and discriminative power of our hallucination metrics beyond aggregate error rates such as WER, using GPT-4o Achiam et al. (2023), we constructed a controlled synthetic dataset designed to elicit specific error types in isolation.[1] This dataset comprises 1,050 synthetic ASR hypothesis-reference pairs across five distinct hallucination categories, 150 per category type: lexical, phonetic, morphological, semantic (local and global), and WER-only divergence, with an additional 150

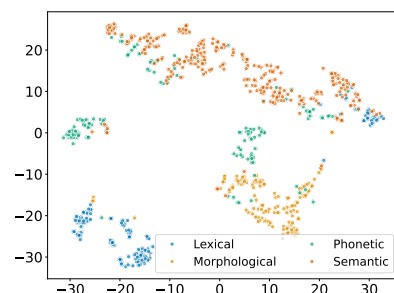

Figure 2: t-SNE projection of SHALLOW metrics, synthetic data.

samples exhibiting mixed error types. As shown in Table 1, each sample was crafted to maximize one hallucination dimension while minimizing confounding signals in others, enabling fine-grained stress testing of the SHALLOW metrics. This synthetic benchmark allows us to empirically demonstrate that the proposed metrics respond in interpretable and non-redundant ways to different hallucination phenomena, capturing distinctions that WER alone cannot resolve. Figure 2 shows a t-SNE projection of SHALLOW metric vectors on such synthetic data, revealing clear separability among hallucination types. Lexical fabrications and morphological errors form compact, distinct clusters, reflecting the precision of their respective metrics. Phonetic fabrications exhibit moderate overlap, due to shared surface-level distortions. Semantic errors are more dispersed, consistent with their broader contextual variability. The structure of the embedding highlights the orthogonality of SHALLOW metrics and supports their effectiveness in disentangling distinct hallucination phenomena beyond WER.

## 2 RELATED WORKS

Hallucination errors are most commonly associated with Natural Language Generation (NLG) and Large Language Models (LLMs), where they involve generating false or fabricated content Huang et al. (2025); Bai et al. (2024). For instance, Martindale et al. (2019) introduced the BVSS metric to detect fluent yet nonsensical outputs using cosine similarity. Most existing NLP hallucination assessments rely on the ROUGE metric, commonly used in summarization Lin (2004). While ROUGE remains widely adopted, it requires access to a parallel corpus. Alternatives like GLEU offer sentence-level fluency evaluation without that constraint Mutton et al. (2007). In Large Vision-Language Models (LVLMs), object hallucination refers to incorrect image descriptions, such as mentioning nonexistent objects or omitting critical elements Zhou et al. (2024).

In ASR, hallucinations arise when neural recognizers generate fluent transcriptions that are unrelated to the input signal, often triggered by ambiguous or noisy speech Barański et al. (2025). This issue stems from the inherent balance between maintaining acoustic fidelity and producing fluent, coherent text. ASR outputs are typically evaluated using error-based metrics like WER, which computes

---

[1] We release the synthetic dataset as part of the SHALLOW benchmark.

the minimum edit distance between reference and hypothesized transcriptions Levenshtein et al. (1966). While easy to calculate, WER has pitfalls, mainly because it assigns an equal penalty to all errors, without providing any insight into the correctness of individual words in the hypothesis Wessel et al. (2002). Those limitations were known before neural ASR models, and speech scientists have proposed several confidence measures, e.g., Wessel et al. (2002); Cox & Rose (1996); Kemp & Schaaf (1997), to label individual words in the ASR output as correct or incorrect, allowing downstream modules to automatically identify potential error locations. However, these measures do not address the fundamental limitations of WER in capturing semantic integrity, differentiating error severity, or handling multiple valid transcriptions Kim et al. (2021b); Mccowan et al. (2004). Alternatives like word information preserved (WIP) Morris et al. (2004) and information retrieval-based metrics Mccowan et al. (2004) address information transfer, while embedding-based metrics focus on semantic similarity Kim et al. (2021b). Yet, none of these metrics adequately capture hallucinations, which represent an emerging, distinct, and problematic class of errors in modern ASR systems based on deep neural networks. As discussed in Atwany et al. (2025), WER can both obscure low hallucination rates and overlook critical and potentially harmful hallucinated content altogether.

The growing use of speech recognition in high-stakes domains like medicine and law, where hallucinated content can lead to severe consequences, underscores the urgent need to address hallucinations in ASR as well. Despite Whisper's overall high accuracy, Koenecke et al. (2024) found that about 1% of its transcriptions included entirely hallucinated phrases not present in the audio. Moreover, 38% of these hallucinations contained explicit harms, such as promoting violence, spreading misinformation, or suggesting false authority. While detailed taxonomies exist for evaluating hallucinations in text generation, such as factual inconsistencies Cattan et al. (2024), knowledge conflicts Xu et al. (2024), and attribution errors Mishra et al. (2024), these frameworks are not applicable to ASR due to its unique challenge: assessing fidelity to an acoustic signal rather than to textual or visual content.

Recent work has begun to properly analyze hallucinations in neural ASR systems. Serai et al. (2022) frames them as (generic) generative errors and explores error prediction using word and phoneme sequences. Frieske & Shi (2024) introduces a perturbation-based approach to evaluate the susceptibility of an ASR model to hallucination at test time using WER, perplexity, and cosine similarity. Barański et al. (2025) presents a refined Bag of Hallucinations method, showing a strong link between hallucinations and training data bias. Finally, Atwany et al. (2025) proposes an LLM-based pipeline using GPT-4o mini to compare ground truth with ASR outputs, categorizing discrepancies into distinct error types, while AssemblyAI Research Team (2024) merely defines hallucinations as N consecutive fabrications.

Prior work on ASR hallucinations such as Frieske & Shi (2024); Barański et al. (2025) remains limited in scope and lacks systematic error categorization. While Atwany et al. (2025) attempts to classify errors, it relies on an LLM that is itself prone to hallucination. SHALLOW addresses this gap by introducing the first benchmarking framework that systematically measures hallucinations across lexical, phonetic, morphological, and semantic dimensions, using targeted metrics to provide interpretable insights into model behavior.

## 3 SHALLOW

In this section, we introduce SHALLOW, the first comprehensive benchmark designed to quantify and categorize ASR hallucinations. While WER treats all errors equally, SHALLOW recognizes that different types of errors vary significantly in their impact on downstream applications and user experience. For instance, a fabricated word that completely changes sentence meaning (e.g., "*not*" inserted before a verb) causes substantially more harm than a morphological variation that preserves semantic intent. Our benchmark employs a taxonomy of hallucinations, systematically evaluating ASR output across four critical dimensions: lexical fabrications (invented content), phonetic fabrications (phonetically similar but lexically incorrect words), morphological errors (structural/grammatical distortions), and semantic errors (semantic inconsistencies within local and global context). Each hallucination channel incorporates multiple weighted components derived from linguistic and computational principles, designed to reflect their relative importance in real-world ASR deployments. The resulting scores offer visibility into ASR model reliability beyond surface-level accuracy metrics.

## 3.1 LEXICAL FABRICATIONS

Lexical fabrications quantify the degree to which the ASR output differs from the reference at the word level. Our analysis distinguishes between three fundamental error types: insertions (added words), substitutions (replaced words), and deletions (removed words). We developed the following composite scoring function that prioritizes insertions as the most damaging form of lexical fabrication:

$$LF = \begin{cases} 1 & \text{if } r_i = 1 \text{ AND } w_i \neq \text{fillers (e.g., 'uhm')} \\ 0.5 \cdot r_i + 0.3 \cdot r_s + 0.2 \cdot r_d & \text{otherwise} \end{cases} \tag{1}$$

Where $r_i$, $r_s$, and $r_d$ represent the ratios of inserted, substituted, and deleted words to total word count, respectively, and $w_i$ the inserted words. The weights (0.5, 0.3, 0.2) reflect empirical observations across our evaluation datasets that insertions typically represent content with minimal acoustic basis in the source audio. At the same time, substitutions maintain structural correspondence, while deletions result in omission rather than the introduction of false information. These weights were validated through analysis of error patterns across diverse domains (detailed in Appendix B). Our framework allows application-specific weight adjustment as needed.

## 3.2 PHONETIC FABRICATIONS

To account for phonetic similarity, which traditional WER ignores, we incorporate three complementary phonetic distance metrics using metaphone transformations Philips (2000) to normalize pronunciation variations. Hamming distance ($H$) captures character-for-character differences, Levenshtein distance ($L$) reflects edit operations needed, and Jaro-Winkler ($JW$) similarity (inverted) accounts for character transpositions and common prefixes. Each metric is normalized to a $[0, 1]$ scale where higher values indicate greater phonetic divergence:

$$PF = \frac{H_N + L_N + (1 - JW)}{3} \tag{2}$$

## 3.3 MORPHOLOGICAL ERRORS

Morphological errors represent distortions in the structural and grammatical properties of ASR output that may preserve core meaning but alter linguistic form. These errors are particularly problematic for applications requiring formal correctness (e.g., educational assessment, transcription services) and for low-resource languages where morphological richness often carries semantic distinctions not present in high-resource languages. Our framework decomposes morphological errors into structural divergence and grammatical errors.

**Structural Divergence.** Structural divergence ($SD$) measures syntactic differences between the reference and the hypothesis at the sentence structure level. It uses dependency parsing to build directed graphs of grammatical relations and computes the (inverse) Jaccard similarity between them. This captures shifts in word relationships or order affecting interpretation.

**Grammatical Errors.** As not all grammatical errors impact comprehension equally, we differentiate error types with a weighted scoring system:

$$GE = \frac{0.4 \cdot E_{Gr} + 0.3 \cdot E_{Sp} + 0.3 \cdot E_{Pu}}{N_{words}} \tag{3}$$

Where $E_{Gr}$, $E_{Sp}$, and $E_{Pu}$ denote grammar, spelling, and punctuation errors, respectively, each normalized by word count.[2] Grammar errors are weighted highest (0.4) as they can completely alter sentence structure, tense, or number agreement, while spelling and punctuation errors carry lower weights (0.3) as they primarily affect formality and clarity without usually changing core meaning.

**Overall Morphological Score.** The composite morphological error score integrates these dimensions with weights reflecting their relative impact:

$$ME = 0.4 \cdot SD + 0.6 \cdot GE \tag{4}$$

Grammatical errors receive the highest weight (0.6) as they most directly impact meaning interpretation. Structural divergence, representing complementary aspects of surface-level linguistic integrity, gets a slightly lower weight (0.4). More details are given in Appendix B.

---

[2]Although punctuation is not produced by all ASR systems, we include punctuation-related cues as they capture structural inconsistencies, such as unclosed clauses or incorrect segmentation, that often surface when punctuation restoration is applied downstream. The list of all possible errors is available at languagetool.org/rules

## 3.4 SEMANTIC ERRORS

While lexical measures capture surface-level changes, semantic error metrics assess how the meaning is preserved regardless of exact wording. These errors are particularly insidious because they may go undetected by traditional metrics yet significantly impact comprehension, especially in longer utterances or conversational speech. This is especially important for ASR systems deployed in critical domains such as healthcare, law, and education. Our framework distinguishes between local (affecting short spans) and global semantic errors (affecting overall meaning coherence).

**Local Semantic Errors.** To detect fine-grained inconsistencies between hypothesis and reference, we define local semantic errors as deviations in meaning observed over short contiguous segments of text. We implement a multi-scale sliding window approach that captures semantic coherence at different granularities. For each window size $w \in \{1, 2, 3\}$, we compute contextual embeddings using a lightweight transformer model Devlin et al. (2019). Each hypothesis window is compared to all reference windows of the same size, retaining the maximum cosine similarity. The coherence score for window $w$ is the average of these maxima, normalized over the longer of the two sequences. The local semantic error score is defined as:

$$LS = 0.5 \cdot (1 - C_1) + 0.3 \cdot (1 - C_2) + 0.2 \cdot (1 - C_3) \tag{5}$$

where $C_1$, $C_2$, and $C_3$ denote semantic alignment for unigrams, bigrams, and trigrams, respectively. Single-token context ($C_1$) receives the highest weight (0.5) to capture word-level semantic shifts, while bi-gram ($C_2$, weight 0.3) and tri-gram ($C_3$, weight 0.2) windows identify phrase-level inconsistencies. This formulation emphasizes token-level distortions while remaining sensitive to higher-order semantic mismatches, effectively detecting cases where individual words maintain semantic plausibility but create contextual dissonance in combination.

**Global Semantic Errors.** Global semantic scores assess semantic coherence across the entire utterance. It includes two metrics, namely semantic distance and semantic coherence.

*Semantic distance.* To compute semantic distance ($SDist$), we first encode the reference and hypothesis sentences using a pretrained sentence embedding model Reimers & Gurevych (2019). We then calculate their cosine similarity, yielding a score in $[0, 1]$ that reflects their alignment in embedding space. Semantic distance is defined as the inverse of this similarity.

*Semantic coherence.* We compute semantic coherence ($SC$) using a hybrid metric that combines BERTScore Zhang et al. (2020) with a contradiction-aware penalty from a natural language inference (NLI) model Lewis et al. (2020). First, we compute the BERTScore F1 between the reference and hypothesis. Then, we classify their semantic relation using a pretrained NLI model. Based on the predicted label, i.e., *entailment*, *neutral*, or *contradiction*, we assign an entailment probability: 1.0, 0.5, or 0.0, respectively. The final coherence score is computed as the product of BERTScore and this probability, yielding high scores only when the hypothesis and reference are both lexically and semantically aligned.

*Overall Global Semantic Score.* The aggregate score combines the components above as follows:

$$GS = \frac{(1 - SDist) + (1 - SC)}{2} \tag{6}$$

Our equal weighting reflects the complementary nature of these dimensions. Preliminary experiments (see Appendix B for more details) confirmed that this balanced approach correlates more strongly with human judgments of hallucination severity than metrics weighted toward either dimension alone.

**Aggregated Semantic Error Score.** To balance fine-grained and holistic semantic evaluation, we compute an aggregated semantic score by linearly combining local and global coherence metrics. For each input pair, we assign a weight of 0.25 to the local semantic score (capturing token- and phrase-level consistency) and 0.75 to the global semantic score (capturing sentence-level meaning preservation). This weighted average prioritizes overall semantic fidelity while still accounting for localized distortions:

$$SE = \frac{1}{4} \cdot LS + \frac{3}{4} \cdot GS \tag{7}$$

## 3.5 SHALLOW EVALUATION FRAMEWORK

We choose not to condense our evaluation into a single composite score. Instead, the SHALLOW benchmark emphasizes reporting the four hallucination dimensions separately:

$$SHALLOW = \{LF, PF, ME, SE\}$$

Table 2: Avg SHALLOW and WER metrics across datasets. Best models in bold (lower is better).

| | HuB | MMS | WLv2 | Canary | WLv3 | Parakeet | SALM. | Q2A | Granite | Kimi | Q2.5O | Phi4 |
|---|---|---|---|---|---|---|---|---|---|---|---|---|
| **WER** | 40.94 | 27.45 | 19.12 | 14.26 | 14.20 | 12.54 | 99.92 | 21.99 | 15.21 | 13.53 | 12.76 | **12.07** |
| **Lexical** | 14.56 | 11.03 | 8.08 | 5.43 | 6.74 | 5.38 | 13.59 | 7.13 | 5.56 | 6.92 | **5.17** | 6.18 |
| **Phonetic** | 35.56 | 26.94 | 20.38 | 16.14 | 17.75 | **15.33** | 27.90 | 21.82 | 15.80 | 20.45 | 16.25 | 17.94 |
| **Morph.** | 27.55 | 23.54 | 13.15 | 11.05 | 11.13 | 10.59 | 16.54 | 13.77 | **10.13** | 12.30 | 10.56 | 11.22 |
| **Semantic** | 35.30 | 26.11 | 17.37 | 14.98 | 14.74 | 13.33 | 23.23 | 19.55 | 13.56 | 15.48 | **12.71** | 14.37 |

Where $LF$ represents lexical fabrications, $PF$ phonetic fabrications, $ME$ denotes morphological errors, and $SE$ semantic errors. This multi-dimensional approach preserves critical information that would be obscured by aggregation, allowing researchers and practitioners to (i) identify specific hallucination patterns in their models without conflating different error types; (ii) track progress on targeted improvements across separate dimensions; (iii) select models based on the hallucination types most relevant to their application domain; and (iv) conduct more nuanced cross-model comparisons beyond simplistic rankings.

## 4 Experimental Setup

Our evaluation covers diverse speech conditions and state-of-the-art ASR architectures, we selected datasets representing various speech challenges and models from diverse architectural paradigms. Complete details on datasets and models are provided in the Appendix A.

**Datasets.** We included multiple categories to test hallucination behavior across different conditions.

*Standard Speech Conditions:* We use LibriSpeech-Other Panayotov et al. (2015) (read audiobooks), TEDLIUM Hernandez et al. (2018) (prepared presentations), and GIGASPEECH Chen et al. (2021) (multi-domain spoken content) to have standardized results on well-studied ASR domains.

*Challenging Acoustic Environments:* CHiME6 Watanabe et al. (2020) provides conversational speech recorded during real dinner parties with natural domestic noise, helping evaluate how environmental challenges may result in different types of hallucinations.

*Heavily-Accented Domains:* We include CORAAL Kendall & Farrington (2023) (African American Language varieties), CV16-Accented Ardila et al. (2020) (accented English), GLOBE-v2 Wang et al. (2024) (164 worldwide English accents), and SpeechOcean Zhang et al. (2021) (non-native English speakers with Mandarin as L1) to evaluate whether accent variation affects hallucination patterns.

*Specialized Domains:* MyST Child Pradhan et al. (2024) includes children's speech in educational contexts, while VoxPopuli Wang et al. (2021) contains formal political speeches, both representing domain-specific vocabulary that may trigger semantic or lexical hallucinations.

**Models.** We evaluated representative models from four distinct ASR architecture families to analyze how architectural choices influence hallucination behaviors.

*Self-Supervised Speech Encoders:* HuBERT (HuB) Hsu et al. (2021) employs masked prediction objectives with a focus on acoustic feature extraction, while MMS Pratap et al. (2024) is a strongly multilingual encoder trained on 1,406 different languages for language-agnostic representation.

*Encoder-Decoder Transformers:* Whisper-Large-v2 (WLv2) and Whisper-Large-v3 (WLv3)Radford et al. (2023a) leverage large-scale weakly supervised training for strong generalization, while Canary Puvvada et al. (2024) uses token-driven decoding for formatting control. This model family balances acoustic and linguistic modeling through specific model sub-networks (e.g., encoder and decoder).

*Encoder-Transducer Models:* We evaluate Parakeet Xu et al. (2023), a FastConformer-based model with monotonic alignment between audio and text sequences. This creates a closer connection between acoustic and linguistic components.

*Multimodal SpeechLLMs:* This newer paradigm includes models that extend linguistic modeling with multimodal speech processing. We include SALMONN (SALM.) Tang et al., Qwen2Audio (Q2A) Chu et al. (2024), Qwen2.5Omni (Q2.5O) Xu et al. (2025), Granite-Speech (Granite) Granite Team (2024), Kimi-Audio (Kimi) Ding et al. (2025), and Phi4-Multimodal-Instruct (Phi4) Abouelenin et al. (2025b). Those models process speech within decoder-only language models, providing a bias towards strong language modeling capabilities.

All models were evaluated using open-source pre-trained weights without domain-specific fine-tuning. The main goal is to assess models' intrinsic hallucination characteristics.

## 5 ANALYSIS OF FINDINGS

Table 2 highlights that SHALLOW metrics expose distinctions across ASR models that WER alone fails to reveal, particularly under diverse acoustic and architectural conditions. While WER favors decoder-only models like Phi4 and Qwen2.5Omni, showing the lowest aggregate error rates, the SHALLOW metrics reveal a more nuanced picture.

For instance, Parakeet achieves the best performance in phonetic and second-best in morphological hallucinations, consistent with its encoder-transducer design that emphasizes acoustic modeling. In contrast, models like Qwen2.5Omni excel in lexical and semantic dimensions, likely due to stronger language modeling components. Interestingly, models with similar WER scores (e.g., Whisper Large-v3 and Canary) exhibit different hallucination profiles: Whisper shows slightly better semantic coherence, while Canary produces fewer lexical fabrications. This validates SHALLOW's ability to disentangle error modalities and expose trade-offs between acoustic fidelity and linguistic fluency. Moreover, the poor alignment between WER and hallucination metrics in models like SALMONN, where WER is extremely high but lexical or semantic scores remain moderate, demonstrates that WER is not predictive of hallucination behavior in low-quality scenarios. This underscores SHALLOW's value in diagnosing failure modes in both state-of-the-art and underperforming models.

Across datasets (see Table 7 in Appendix D.2), we observe dataset-specific sensitivity. For example, in CORAAL, hallucination metrics rise across the board, with SpeechLLMs suffering more than other models, reflecting linguistic mismatch and highlighting the need for inclusive acoustic-linguistic modeling. In CHiME6, phonetic hallucination scores are uniformly high across all models. This indicates that conversational overlap and acoustic degradation pose a persistent challenge to phoneme-level decoding, independent of overall WER. SHALLOW makes this failure mode visible even when aggregate metrics suggest acceptable performance. Conversely, in simpler datasets like Librispeech, hallucination scores are consistently low, suggesting that SHALLOW metrics align with expected difficulty variations, reinforcing their interpretive value.

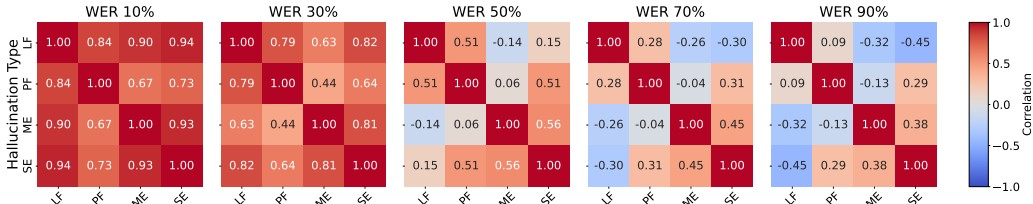

Figure 3: Spearman Correlation of SHALLOW metrics across different WER values.

**Metrics correlation.** Figure 3 reports the Spearman correlation between SHALLOW's four hallucination metrics across five WER regimes. At low WER (10–30%), the metrics exhibit strong monotonic relationships ($\rho_s > 0.80$), indicating that when errors are sparse, they tend to co-occur and scale similarly across categories. However, as WER increases, these correlations degrade substantially. By WER 50%, several metric pairs show weak or even negative associations (e.g., Lexical–Morphological at $\rho_s = -0.14$), and by WER 90%, correlations such as Lexical–Semantic drop below –0.45. This trend suggests that hallucination types decouple under degraded conditions: models may produce syntactically fluent but semantically implausible outputs, or preserve meaning while distorting surface forms. These findings validate SHALLOW's design as a multidimensional lens for ASR evaluation. Whereas WER obscures the nature and structure of errors, especially under failure-prone conditions, SHALLOW's metrics remain discriminative and interpretable, revealing distinct error behaviors that emerge as recognition quality deteriorates. This makes SHALLOW particularly valuable in low-resource, noisy, or out-of-domain settings, where models' hallucination profiles can diverge sharply despite similar overall error rates.

**Case study on downstream task: Medical ASR.** To demonstrate SHALLOW's practical value in critical domains, we conducted a zero-shot analysis of Phi4 ASR performance in medical settings, where transcription errors can directly impact patient care. Using the Medical-ASR[3] and AfriSpeech Olatunji et al. (2023) (clinical domain) datasets, we identified several cases where WER fails to capture potentially dangerous errors. Results are shown in Table 3. For instance, when

---

[3] https://huggingface.co/datasets/jarvisx17/Medical-ASR-EN

Table 3: Clinical speech recognition error analysis for Phi4 model, SHALLOW metrics.

| Reference | Hypothesis | WER | LF | PF | ME | SE |
|---|---|---|---|---|---|---|
| Medical-ASR Dataset | | | | | | |
| i can not rotate my neck | i can rotate my neck | 0.16 | 3.33 | 29.36 | 6.67 | 60.79 |
| i feel like the room is spinning | i feel like the room is empty | 0.14 | 4.29 | 18.14 | 29.09 | 56.75 |
| is my cut infected or just healing | is my cat infected or just healing | 0.14 | 4.29 | 0.00 | 17.78 | 56.27 |
| i have a problem in my back i cannot extend it | i have a problem in my bag i cannot stand it | 0.18 | 5.45 | 9.63 | 29.47 | 60.46 |
| it is hard to see things | it is hard to say things | 0.17 | 5.00 | 0.00 | 26.67 | 60.11 |
| i feel pain in my knee | i feel pain in my neck | 0.17 | 5.00 | 9.33 | 26.67 | 51.55 |
| i feel lightheaded | i feel light headed | 0.67 | 22.50 | 26.80 | 27.00 | 8.51 |
| i cant breathe | i can not breathe | 0.67 | 22.50 | 29.22 | 26.67 | 11.16 |
| red flushes accompanied with itchy | red flush is accompanied with itching | 0.60 | 20.33 | 33.64 | 36.00 | 13.27 |
| AfriSpeech Dataset (Clinical Domain) | | | | | | |
| the ulna remains relatively stationary | the owner remains relatively stationary | 0.20 | 6.00 | 12.48 | 22.86 | 37.63 |
| took 62 and 35 cc well with yellow nipple | took 62 and 335 cc well with yellow nipple | 0.11 | 3.33 | 0.00 | 8.00 | 42.44 |
| reason bilat pe eval for dvt | reason bilateral p e evaluation for dvt | 0.67 | 22.00 | 32.02 | 42.67 | 19.38 |

transcribing "*I can not rotate my neck*" as "*I can rotate my neck*", the model produces a critical polarity flip with a falsely low WER of 0.16. While WER and LF (3.33) suggest minor deviation, SHALLOW's high SE score (60.79) correctly flags this as a severe error that inverts the patient's reported symptom. Similarly, transcribing "*I feel like the room is spinning*" as "*I feel like the room is empty*" replaces a clear indicator of vertigo with an unrelated description. Despite a low WER (0.14), SHALLOW's high SE score (56.75) appropriately identifies the loss of crucial diagnostic information. Moreover, even single-letter substitutions can be critical: changing "*cut*" to "*cat*" in a query about infection (WER = 0.14) completely alters the medical context, which SHALLOW captures through elevated SE (56.27) despite low PF scores. In the AfriSpeech dataset, transcribing "*the ulna remains relatively stationary*" as "*the owner remains relatively stationary*" demonstrates how phonetically plausible errors (PF = 12.48) can still produce nonsensical medical observations, correctly captured by SHALLOW's SE score (37.63) despite a low WER (0.20). These examples highlight SHALLOW's ability to identify potentially harmful transcription errors that traditional metrics might miss, making it particularly valuable for evaluating ASR systems in healthcare applications.

# 6 CONCLUSIONS

We have introduced SHALLOW, the first comprehensive benchmark for characterizing and quantifying hallucinations in ASR systems. By decomposing ASR errors into four complementary dimensions, i.e., lexical, phonetic, morphological, and semantic, SHALLOW provides interpretable profiles of model behavior that conventional metrics like WER fail to capture. Our evaluation across diverse architectures and domains demonstrates that hallucination patterns vary significantly based on model design choices and acoustic conditions, with divergence from WER scores in challenging scenarios. The consistent breakdown of correlation between SHALLOW metrics and WER as recognition quality degrades validates the framework's ability to identify fine-grained error structure that would otherwise remain obscured. SHALLOW allows to diagnose specific model weaknesses, select systems based on application-specific requirements, and measure targeted progress beyond aggregate accuracy scores.

**Limitations.** SHALLOW deliberately provides four distinct scores representing our hallucination dimensions, each computed as a weighted aggregate of several component metrics rather than reporting all individual sub-metrics separately. While this approach offers interpretable profiles along our four primary axes, the assigned weights necessarily reflect an assessment of relative importance that cannot be universally optimal across all ASR applications and domains. Accessing individual scores is still possible but would limit interpretability and actionable insights. Our framework currently focuses on English evaluation, with particular constraints in the semantic error dimension, which relies on language-specific NLP models and contextual embeddings. SHALLOW can be readily extended to other languages, provided that semantically-rich embedding models are available. This reflects a more broad constraints in multilingual NLP for hallucination evaluation.

# 7 ETHICS STATEMENT

This work aims to improve ASR safety by providing better tools for detecting hallucinations, particularly in critical applications like healthcare and legal transcription where errors can cause harm. Our

evaluation includes datasets representing diverse speech varieties (CORAAL, accented English) to ensure inclusive assessment, though we acknowledge that benchmarking on dialectal speech carries risks if results are misinterpreted to suggest deficiencies in certain speech communities rather than model limitations. We emphasize that SHALLOW is designed to diagnose model weaknesses for improvement, not to rank speech varieties, and encourage responsible use that promotes equitable ASR development across all user populations. The medical case studies demonstrate potential harms from ASR hallucinations but are presented solely to motivate better evaluation practices, not to discourage ASR deployment in healthcare where benefits may outweigh risks when proper safeguards are implemented.

## 8    REPRODUCIBILITY STATEMENT

To ensure full reproducibility, we provide comprehensive implementation details for all SHALLOW metrics in the appendix, including specific libraries (Appendix C), computational procedures and edge-case handling (Appendix F), and the link to our open-source framework.[4] Our synthetic validation dataset of 1,050 hypothesis-reference pairs is released alongside the complete SHALLOW framework code, and extensively described in Appendix B. All evaluated models use publicly available checkpoints with exact version specifications provided in Table 5 (Appendix A), and dataset processing details are documented in Appendix A, Table 4. The modular design of our framework enables straightforward extension to new models and datasets, with all hyperparameters and weighting schemes explicitly specified in Section 3 and Appendix C.

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

# A  DATASETS AND MODELS DETAILS

This appendix section provides complete information about the datasets and models used in our experimental evaluation of the SHALLOW benchmark framework. Tables 4 and 5 summarize the key characteristics of the datasets and models, respectively.

Table 4: Summary of datasets used in the SHALLOW benchmark evaluation.

| Dataset | # Test Utts | Domain | Characteristics |
|---|---|---|---|
| *Standard Speech Conditions* | | | |
| LibriSpeech (other) Panayotov et al. (2015) | 2,939 | Read audiobooks | Standard "other" split with more challenging samples |
| TEDLIUM Hernandez et al. (2018) | 1,469 | TED talks | Clear, prepared speech by professional speakers |
| GIGASPEECH Chen et al. (2021) | 25,619 | Diverse sources | Audiobooks, podcasts, YouTube; diverse topics |
| *Challenging Acoustic Environments* | | | |
| CHiME-6 Watanabe et al. (2020) | 11,027 | Dinner parties | Conversational speech with natural domestic noise |
| *Heavily-Accented Domains* | | | |
| CORAAL Kendall & Farrington (2023) | 5,000 | Interview speech | Regional varieties of African American Language |
| CV16-Accented Ardila et al. (2020) | 2,197 | Crowd-sourced | English utterances with accent variation |
| GLOBE-v2 Wang et al. (2024) | 5,046 | Global accents | 164 accents from worldwide speakers |
| SpeechOcean Zhang et al. (2021) | 2,500 | L2 English | Non-native speakers (L1: Mandarin); children and adults |
| *Specialized Domains and Voices* | | | |
| MyST Child Pradhan et al. (2024) | 13,180 | Educational | Children (grades 3-5) with virtual science tutor |
| VoxPopuli Wang et al. (2021) | 1,842 | Political speeches | Formal speaking with domain-specific terminology |

## A.1  DATASETS

We selected datasets representing diverse speech conditions, domains, and challenges that ASR systems encounter in real-world applications. The following statistics describe the test sets of the respective datasets.

**Standard Speech Conditions:** LibriSpeech (other) Panayotov et al. (2015) contains 2,939 test utterances from read audiobooks that typically yield low WER scores across modern systems. We use the standard "other" split, which includes more challenging speech samples than the "clean" split. TEDLIUM Hernandez et al. (2018) includes 1,469 test utterances from English-language TED talks, representing clear, prepared speech in a presentation setting with professional speakers. GIGASPEECH Chen et al. (2021) comprises 25,619 test utterances from a multi-domain corpus spanning audiobooks, podcasts, and YouTube videos, covering both read and spontaneous speech across diverse topics including arts, science, and sports, with high-quality transcriptions.

**Challenging Acoustic Environments:** CHiME-6 Watanabe et al. (2020) includes 11,027 test utterances recorded during real dinner parties in everyday home environments. This dataset captures conversational speech with natural domestic noise from kitchen appliances, air conditioning, and movement across various room acoustics.

**Heavily-Accented Domains:** CORAAL Kendall & Farrington (2023) contains utterances from the Corpus of Regional African American Language, sampled from sociolinguistic interviews representing regional varieties of African American Language. It includes audio recordings with time-aligned transcriptions. We selected a subset of 5,000 test samples. CV16-Accented Ardila et al. (2020) consists of 2,197 test utterances from the CommonVoice corpus, specifically selected as English utterances labeled with accent variation. GLOBE-v2 Wang et al. (2024) provides 5,046 test utterances with worldwide English accents, covering 164 accents from over 23,000 speakers, making it ideal for testing accent generalization. SpeechOcean Zhang et al. (2021) includes 2,500 test utterances from non-native English speakers whose first language is Mandarin, with balanced data from both children and adults with expert-scored pronunciations.

**Specialized Domains and Voices:** MyST Child Pradhan et al. (2024) includes 13,180 test utterances with transcription from children in grades 3-5 conversing with a virtual science tutor, combining children's speech patterns with scientific vocabulary in educational applications. VoxPopuli Wang et al. (2021) contains 1,842 test utterances from political speeches, offering transcribed formal speaking styles with domain-specific terminology.

Table 5: Summary of ASR models evaluated in the SHALLOW benchmark.

| Model | Architecture Type | # Params | Key Characteristics |
|---|---|---|---|
| *Self-Supervised Speech Encoders* | | | |
| HuBERT Hsu et al. (2021) | Encoder-only | 300M | Masked prediction objectives; fine-tuned on LibriSpeech |
| MMS Pratap et al. (2024) | Encoder-only | 1B | Multilingual (1,406 languages); language-agnostic representations |
| *Encoder-Decoder Transformers* | | | |
| Whisper-Large-v2 Radford et al. (2023a) | Encoder-decoder | 1.5B | 680,000 hours of weakly supervised multilingual training |
| Whisper-Large-v3 | Encoder-decoder | 1.5B | 5M+ hours training data; enhanced generalization capabilities |
| Canary Puvvada et al. (2024) | Encoder-decoder | 1B | FastConformer encoder (32 layers); token-driven decoding |
| *Encoder-Transducer Models* | | | |
| Parakeet Xu et al. (2023) | Encoder-transducer | 1.1B | FastConformer-based; optimized for English recognition |
| *Multimodal SpeechLLMs* | | | |
| SALMONN Tang et al. | Decoder w/ encoders | 7B | Integrates LLMs with speech/audio encoders; unified processing |
| Qwen2Audio Chu et al. (2024) | Decoder w/ encoders | 8.4B | Part of Qwen2 series; specialized audio encoders |
| Qwen2.5-Omni Xu et al. (2025) | Decoder w/ encoders | 10.7B | Enhanced Qwen2; broader multimodal capabilities |
| Granite-Speech Granite Team (2024) | Decoder w/ encoders | 8.6B | Two-pass design for transcription and translation |
| Kimi-Audio Ding et al. (2025) | Decoder w/ encoders | 9.7B | Open audio model; unified framework for audio tasks |
| Phi4-MM-Instruct Abouelenin et al. (2025b) | Decoder w/ encoders | 5.6B | Open-weights foundation model; Multimodal by design. |

## A.2 MODELS

We evaluated representative models from four distinct ASR architecture families, each employing different approaches to speech processing.

**Self-Supervised Speech Encoders:** HuBERT[5] Hsu et al. (2021) is a self-supervised model trained on masked prediction objectives and fine-tuned on 960 hours of LibriSpeech data. It uses discrete speech units learned through iterative clustering and has demonstrated strong performance on several downstream speech tasks. MMS[6] Pratap et al. (2024) is a multilingual speech encoder based on the wav2vec 2.0 architecture Baevski et al. (2020), trained on 1,406 languages. Unlike language-specific models, MMS extracts language-agnostic representations that aim to generalize across linguistic patterns. Encoder-only models typically focus on acoustic fidelity and may struggle in generating linguistically coherent outputs, potentially impacting morphological and semantic hallucination metrics.

**Encoder-Decoder Transformers:** Whisper-Large-v2[7] Radford et al. (2023a) is an encoder-decoder transformer trained on 680,000 hours of weakly supervised multilingual data, demonstrating impressive zero-shot generalization across diverse domains and acoustic conditions. Whisper-Large-v3[8] is an enhanced version trained on over 5 million hours of data, maintaining the architecture of its predecessor with refinements to enhance generalization capabilities. Canary[9] Puvvada et al. (2024) is a specialized encoder-decoder model with a FastConformer encoder (32 layers) and a transformer decoder (4 layers), comprising approximately 883M parameters. This model uses token-driven decoding for controlling transcription format, timestamps, and multilingual capabilities. Encoder-decoder models balance acoustic and linguistic modeling, potentially showing more controlled hallucination patterns across multiple dimensions compared to other architectural families.

---

[5] https://huggingface.co/facebook/hubert-large-ls960-ft
[6] https://huggingface.co/facebook/mms-1b-all
[7] https://huggingface.co/openai/whisper-large-v2
[8] https://huggingface.co/openai/whisper-large-v3
[9] https://huggingface.co/nvidia/canary-1b-flash

**Encoder-Transducer Models:** Parakeet[10] Xu et al. (2023) is a FastConformer-based encoder-transducer model optimized for English speech recognition. Transducers employ monotonic alignment between audio and text, potentially influencing their hallucination patterns in continuous speech. The joint network creates tighter coupling between acoustic and linguistic components, which may yield distinct hallucination behavior compared to more loosely coupled encoder-decoder systems.

**Multimodal SpeechLLMs:** SALMONN[11] Tang et al. integrates pre-trained text-based LLMs with speech and audio encoders, processing speech, audio events, and music within a unified framework. Qwen2Audio[12] Chu et al. (2024) is part of the Qwen2 series, with the decoder-only LLM processing audio signals through specialized encoders before generating text responses. We also evaluated Qwen2.5-Omni[13] Xu et al. (2025), which support broader multimodal capabilities. Granite-Speech[14] Granite Team (2024) is a compact decoder-only model employing a two-pass design for transcribing and translating audio inputs. Kimi-Audio[15] Ding et al. (2025) is an open audio model supporting a range of audio processing tasks (including ASR) within a single unified framework. Phi4-Multimodal-Instruct[16] Abouelenin et al. (2025b) is an open-weights multimodal foundation model that processes speech inputs alongside text and images. It shows state-of-the-art performance on ASR task. Decoder-only models have stronger language modeling capabilities, which may result in more fluent outputs but potentially higher phonetic or lexical hallucinations due to stronger linguistic priors.

All models were evaluated using authors-provided pre-trained weights without domain-specific fine-tuning to assess their intrinsic hallucination characteristics.

## B  SYNTHETIC BENCHMARK DATASET

To rigorously evaluate the SHALLOW metrics under controlled conditions, we introduce a synthetic benchmark dataset designed to isolate individual types of hallucination phenomena in ASR transcriptions. This dataset enables precise analysis of how each metric responds to specific, targeted perturbations, which would be difficult to disentangle in naturally occurring ASR errors.

### B.1  MOTIVATION

While real-world speech corpora are essential for measuring end-to-end ASR performance, they often contain entangled sources of error, i.e., acoustic noise, disfluencies, dialectal variation, and domain mismatch, making it difficult to attribute hallucination metrics to specific error types. In proposing the SHALLOW framework, we wanted to isolate individual hallucination phenomena to validate each metric responds specifically to its intended error category. Aggregate measures like WER offer no insight into the structure of such errors. In contrast, a synthetic dataset allows us to test metric behavior under clean, deliberately controlled conditions where individual hallucination categories are introduced in isolation.

This enables fine-grained stress testing and validation of key metric properties: interpretability, orthogonality, and semantic sensitivity, particularly in edge cases where WER alone fails.

### B.2  DATASET COMPOSITION

The dataset consists of 1,050 synthetic hypothesis–reference pairs, evenly distributed across six hallucination categories:

- *Lexical Fabrication (150)*: Fluent hallucinations introducing unrelated content not present in the reference.
- *Phonetic Confusion (150)*: Substitutions involving phonetically similar but incorrect words (e.g., "there" vs. "their").

---

[10]https://huggingface.co/nvidia/parakeet-rnnt-1.1b
[11]https://huggingface.co/tsinghua-ee/SALMONN-7B
[12]https://huggingface.co/Qwen/Qwen2-Audio-7B
[13]https://huggingface.co/Qwen/Qwen2.5-Omni-7B
[14]https://huggingface.co/ibm-granite/granite-speech-3.3-8b
[15]https://huggingface.co/moonshotai/Kimi-Audio-7B-Instruct
[16]https://huggingface.co/microsoft/Phi-4-multimodal-instruct

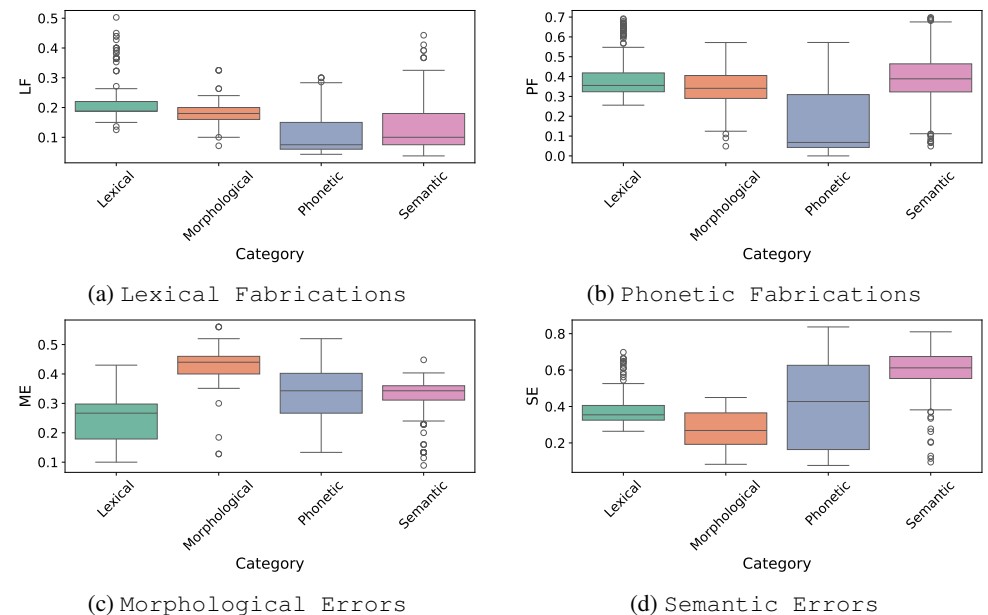

(a) Lexical Fabrications

(b) Phonetic Fabrications

(c) Morphological Errors

(d) Semantic Errors

Figure 4: Distribution of hallucination scores across categories for each SHALLOW metric. Each subplot shows box plots of metric values per hallucination type. For most metrics, scores peak in their intended category. The `PF` metric is lowest on phonetic samples, reflecting successful detection of phonetic proximity.

- *Morphological Divergence (150)*: Grammatical or punctuation-level distortions (e.g., verb tense, agreement, or sentence boundaries).
- *Semantic Drift (300)*: Shifts in meaning, including polarity reversals or role inversion, while preserving lexical fluency. Includes both local (150) and global (150) variants.
- *WER-only Divergence (150)*: High surface-level WER but semantically equivalent hypotheses (e.g., paraphrased or reordered content).
- *Mixed Errors (150)*: Hypotheses with multiple overlapping hallucination types, reflecting realistic multi-dimensional failures.

Each reference is a short, unambiguous sentence in standard English. Hypotheses are generated using GPT-4o Achiam et al. (2023) under type-specific prompts to maximize the intended error while minimizing confounding factors. This construction supports precise validation of metric performance on the phenomena they are meant to detect.

### B.3 GENERATION METHODOLOGY

We used GPT-4o to generate synthetic hypotheses from handcrafted references, using structured prompting tailored to each hallucination category. For example:

- For phonetic confusions, we employed a metaphone-based similarity filter to replace content words with phonetically similar alternatives.
- For semantic drift, we prompted the model to alter meaning without obvious lexical deviation, ensuring plausibility and fluency.
- Morphological errors were crafted by introducing subject-verb agreement errors or incorrect tenses.
- WER-only examples involved paraphrasing references such that WER increases while meaning is preserved, stressing the metric's discriminative capacity.

Each pair was manually reviewed to ensure alignment with the intended category and avoid noise from model hallucination or overlap.

Table 6: Examples of synthetic data with WER and all SHALLOW metrics. Each block focuses on different error categories. $\mathbf{r}_i$, $\mathbf{r}_d$. $\mathbf{r}_s$ report the insertion, deletion, and substitution ratio, respectively. $\mathbf{H}_N$, $\mathbf{L}_N$, and $\mathbf{1} - \mathbf{JW}$ indicate the Hamming distance (normalized), the Levenshtein distance (normalized), and the inverse of the Jaro-Winkler similarity. $\mathbf{SD}$ denotes the structural divergence, while $\mathbf{E}_{Gr}$, $\mathbf{E}_{Sp}$, and $\mathbf{E}_{Pu}$ are the grammar, spelling, and punctuation errors, respectively, which sum up to the grammatical errors $\mathbf{GE}$. $\mathbf{L}_{w1}$, $\mathbf{L}_{w2}$, and $\mathbf{L}_{w3}$ mark the local semantic scores for windows considering 1, 2, and 3 words, respectively. $\mathbf{SDist}$ stands for semantic distance, while $\mathbf{1} - \mathbf{SC}$ indicates the inverse of the semantic coherence. $\mathbf{LF}$, $\mathbf{PF}$, $\mathbf{ME}$, and $\mathbf{SE}$ denote the aggregate scores for lexical, phonetic, morphological, and semantic categories, respectively.

| | Reference | Hypothesis | WER | $\mathbf{r}_i$ | $\mathbf{r}_d$ | $\mathbf{r}_s$ | | LF | |
|---|---|---|---|---|---|---|---|---|---|
| Lexical | She left her keys at home | She forgot her keys | 0.50 | 0.00 | 0.33 | 0.17 | | 0.12 | |
| | We watched the sun set at the beach | We screamed the sun set at the beach and danced | 0.38 | 0.20 | 0.00 | 0.13 | | 0.14 | |
| | She opened a window | She breached the wall portal to let space in | 2.00 | 0.56 | 0.00 | 0.75 | | 0.50 | |

| | Reference | Hypothesis | WER | $\mathbf{H}_N$ | $\mathbf{L}_N$ | $\mathbf{1} - \mathbf{JW}$ | | PF | |
|---|---|---|---|---|---|---|---|---|---|
| Phonetic | She bakes with flour | She baks with flower | 0.50 | 0.15 | 0.07 | 0.02 | | 0.08 | |
| | I cleaned the kitchen | I leaned the kitchen | 0.25 | 0.83 | 0.08 | 0.02 | | 0.31 | |
| | I will buy it for you | Isle by it 4 ewe | 0.83 | 0.93 | 0.43 | 0.36 | | 0.57 | |

| | Reference | Hypothesis | WER | SD | $\mathbf{E}_{Gr}$ | $\mathbf{E}_{Sp}$ | $\mathbf{E}_{Pu}$ | GE | ME |
|---|---|---|---|---|---|---|---|---|---|
| Morph. | We enjoy watvching birds | We enjoy watching birds frequentlier | 0.25 | 0.20 | 0.00 | 1.00 | 0.00 | 0.08 | 0.13 |
| | He painted the wall red | He paints walls redly | 0.80 | 1.00 | 0.00 | 0.00 | 0.00 | 0.00 | 0.40 |
| | They ride horses | They rided horses quickierly | 0.67 | 1.00 | 2.00 | 0.00 | 0.00 | 0.20 | 0.52 |

| | Reference | Hypothesis | WER | $\mathbf{L}_{w1}$ | $\mathbf{L}_{w2}$ | $\mathbf{L}_{w3}$ | SDist | $\mathbf{1} - \mathbf{SC}$ | SE |
|---|---|---|---|---|---|---|---|---|---|
| Semantic | I picked a red flower | I picked a dead flower | 0.20 | 0.96 | 0.71 | 0.54 | 0.40 | 0.77 | 0.46 |
| | The big house is old | The small house is new | 0.40 | 0.94 | 0.73 | 0.52 | 0.64 | 1.00 | 0.67 |
| | He played video games | He fought sports | 0.75 | 0.65 | 0.34 | 0.18 | 0.71 | 1.00 | 0.77 |

## B.4 EXAMPLES

Table 6 shows some representative examples from the synthetic benchmark, illustrating how different error types are instantiated.

## B.5 METRIC DISTRIBUTION ON THE SYNTHETIC DATASET

Figure 4 presents the distribution of SHALLOW metric scores across synthetic samples, grouped by their intended hallucination category. Each subplot shows a box plot for one metric (Lexical Fabrications `LF`, Phonetic Fabrications `PF`, Morphological Errors `ME`, Semantic Errors `SE`) computed over the synthetic pairs stratified by hallucination type (Lexical, Morphological, Phonetic, Semantic).

The goal of this analysis is to validate the specificity and discriminative capacity of each SHALLOW metric: ideally, a given metric should produce the highest values for samples in its target category, while assigning relatively low scores to samples from other categories. This behavior would confirm that the metrics are aligned with their intended error modalities and are not conflating unrelated phenomena.

**Lexical Fabrication (a):** As expected, the `LF` metric exhibits the highest values for samples in the `Lexical` category, indicating that these hypotheses introduce content absent from the reference. Other categories yield lower scores, with the median sharply reduced, consistent with the dataset's design to minimize lexical novelty outside the intended axis.

**Phonetic Fabrication (b):** Unlike the other panels, the `PF` metric shows an inverted pattern: the `Phonetic` category has the *lowest* median score. This is by design. In this benchmark, phonetic hallucination samples were generated by introducing phonetically plausible substitutions (e.g., "*there*" → "*their*"), which should yield low phonetic distance if the metric works correctly. Thus, `PF` scores

being minimized here is a positive validation: it confirms that the metric detects phonetic proximity rather than penalizing substitutions indiscriminately.

**Morphological Errors (c):**   The `ME` metric peaks in the `Morphological` category, as intended. These errors often involve tense, number, and overall sentence structure (e.g., "The cat run" vs. "The cat runs."), which are designed to specifically challenge grammatical and structural consistency. Other categories display modest scores, affirming metric specificity.

**Semantic Errors (d):**   The `SE` metric exhibits highest median scores for the `Semantic` category, capturing both local and global meaning shifts. While samples from other categories may contain some incidental semantic variation, their scores remain clearly lower, validating the semantic isolation in the dataset construction.

Taken together, these distributions empirically confirm that SHALLOW metrics react most strongly to their corresponding hallucination types and remain relatively unaffected by unrelated errors. This demonstrates both the targeted design quality of the synthetic benchmark and the functional separability of SHALLOW metrics, which is crucial for their use in detailed ASR hallucination diagnostics. This synthetic dataset thus plays a fundamental role in validating SHALLOW by allowing: (i) *Metric specificity testing*, ensuring each metric responds only to its target error category; (ii) *Correlation analysis*, demonstrating low inter-metric correlation in isolated conditions; (iii) *Controlled counterexamples*, stress-testing metrics on adversarial or benign WER-only cases. This benchmark is released as part of the SHALLOW framework[17] to facilitate reproducibility, benchmarking, and future research into fine-grained ASR hallucination detection.

### B.6   Spearman Correlation Analysis

Figure 5 shows the Spearman correlation matrix computed over the synthetic dataset, assessing the relationships between each SHALLOW metric and WER. As expected, the Lexical Fabrications (LF) metric exhibits an almost perfect correlation with WER ($\rho = 0.98$), confirming that lexical insertions and substitutions are the primary drivers of overall word-level mismatch in most ASR hallucinations. Phonetic Fabrications (PF) and Morphological Errors (ME) show moderate positive correlations with WER ($\rho = 0.54$ and $0.51$, respectively), suggesting that these dimensions contribute to error accumulation but are not always aligned with aggregate WER changes. Semantic Errors (SE) are only weakly correlated with WER ($\rho = 0.15$), reinforcing

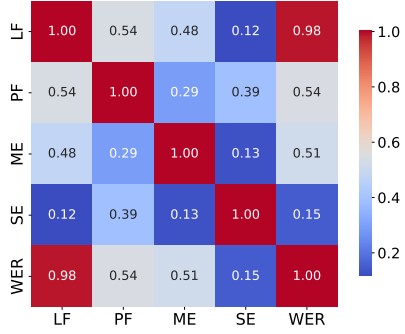

Figure 5: Spearman correlation of hallucination scores, synthetic data.

the idea that semantically misleading outputs can occur even when WER is low, and vice versa. The low correlations between SE and other metrics (e.g., LF–SE: 0.12, ME–SE: 0.13) further highlight the orthogonality of semantic hallucinations within the SHALLOW framework. These findings support our core claim: SHALLOW captures complementary error dimensions that WER alone fails to distinguish, particularly in cases where fluency masks semantic distortion.

## C   Metric Implementation Details

This section provides implementation-specific details for the SHALLOW metrics described in Section 3. We focus on computational considerations, optimizations, and technical choices that complement the theoretical framework presented in the main paper.

### C.1   Lexical Fabrication Metrics

The lexical fabrication metrics quantify word-level deviations between reference and hypothesis transcripts. We implement these metrics using the `JiWER` library[18] to compute insertions, deletions,

---

[17]See: `https://anonymous.4open.science/r/SHALLOW/`
[18]`https://github.com/jitsi/jiwer`

and substitutions between transcription pairs. For each reference-hypothesis pair, we calculate the relative ratios of these error types. Insertion ratio is computed as the number of inserted words divided by the total word count in the hypothesis. Deletion ratio represents removed words relative to the reference length. Substitution ratio captures replaced words as a proportion of reference length.

Special handling is implemented for edge cases, including empty references or hypotheses.

1: **if** reference = hypothesis **then**
2:     **return** $\{ins = 0, del = 0, sub = 0\}$          ▷ Short-circuit for exact matches
3: **else if** len(reference) = 0 **then**
4:     **return** $\{ins = |hypothesis|, ins\_ratio = 1.0, del = 0, sub = 0\}$
5: **else if** len(hypothesis) = 0 **then**
6:     **return** $\{ins = 0, del = |reference|, del\_ratio = 1.0, sub = 0\}$
7: **end if**

Our implementation detects and excludes common speech disfluencies (e.g., "*um*," "*uh*") from the insertion count when applying the final weighting formula, as these are considered standard elements of conversational speech rather than hallucinations.

## C.2 PHONETIC FABRICATION METRICS

Phonetic fabrication metrics evaluate the degree of phonetic dissimilarity between reference and hypothesis transcriptions. Our implementation leverages the `Jellyfish` library[19] to transform textual content into metaphone representations, which normalize pronunciation variations. This phonetic encoding converts words to approximate phonetic equivalents, enabling comparison based on pronunciation rather than spelling. We compute three complementary phonetic distance metrics between the metaphone-encoded reference and hypothesis:

1. *Hamming distance*: Measures character-for-character differences, normalized by the length of the longer string between the reference and hypothesis.

2. *Levenshtein distance*: Quantifies the minimum number of single-character edits (insertions, deletions, substitutions) required to transform one string into another, also normalized by the maximum string length.

3. *Jaro-Winkler similarity*: Captures character transpositions and common prefixes, returning a similarity score between 0 and 1.

All distance metrics are normalized to the $[0, 1]$ range using the maximum possible distance (i.e., the longer string length) rather than using absolute values, enabling consistent scaling across utterances of different lengths. The combined score (as described in Section 3.2) provides a robust measure of phonetic discrepancy that accounts for different aspects of pronunciation variation.

## C.3 MORPHOLOGICAL ERROR METRICS

Morphological error metrics assess structural and grammatical distortions in ASR outputs. Our implementation combines syntax tree comparison with grammar checking to evaluate how ASR systems preserve linguistic structure.

For structural analysis, we use `SpaCy` Honnibal et al. (2020) with the Berkley neural constituency parser Kitaev & Klein (2018)[20] to build dependency trees for both reference and hypothesis texts. Each sentence is represented as a set of dependency relations in the form of (head, dependency relation, token) triples. We compute structural divergence using the Jaccard distance between the reference and hypothesis dependency sets:

$$SD = 1 - \frac{|R \cap H|}{|R \cup H|} \tag{8}$$

where $R$ and $H$ represent the sets of dependency relations for reference and hypothesis, respectively. This metric captures differences in grammatical relationships and word order that may affect interpretation.

---

[19] https://github.com/jamesturk/jellyfish
[20] https://github.com/nikitakit/self-attentive-parser

For grammatical error analysis, we employ the `LanguageTool API`[21] to detect and categorize errors in the hypothesis text. Errors are classified into three primary categories (e.g., Grammar, Spelling, and Punctuation errors) and aggregated using a specific weighting scheme as described in the main manuscript. The final morphological error score integrates both structural and grammatical error analysis into a final score as described in Section 3.3.

### C.4 SEMANTIC ERROR METRICS

Semantic error metrics evaluate the preservation of meaning between reference and hypothesis transcriptions. Our implementation distinguishes between local semantic errors (affecting short spans) and global semantic coherence (affecting overall meaning).

For local semantic analysis, we employ a multi-scale sliding window approach using contextual embeddings from BERT-based models Devlin et al. (2019). For each window size $w \in \{1, 2, 3\}$ (unigrams, bigrams, trigrams), we:

1. Compute contextual embeddings for each window in both the reference and the hypothesis;

2. Compare each hypothesis window to all reference windows of the same size using cosine similarity;

3. Retain the maximum similarity score for each hypothesis window;

4. Average these maximum scores, normalized by the length of the longer sequence.

The local semantic error score is computed using a weighted scheme for different window sizes as described in Section 3.4.

For global semantic analysis, we compute two complementary metrics:

1. *Semantic distance* ($SDist$): Computed as the inverse of cosine similarity between sentence-level embeddings generated by a RoBERTA-based model Liu et al. (2019) optimized for NLI tasks.[22]

2. *Semantic coherence* ($SC$): Combines BERTScore F1 with a contradiction-aware penalty from a BART-based Lewis et al. (2020) natural language inference (NLI) model.[23]

Extending previous work on the importance of the semantic dimension in ASR evaluation Kim et al. (2021a), our semantic coherence score integrates NLI predictions by scaling the BERTScore with an entailment probability factor:

- 1.0 for entailment classification (reference entails hypothesis)

- 0.5 for neutral classification (no clear relationship)

- 0.0 for contradiction classification (reference contradicts hypothesis)

The global semantic error score averages these components and the final semantic error score combines local and global components with a 1:3 ratio.

## D COMPREHENSIVE RESULTS ACROSS SPEECH DATASETS

Table 7 reports WER and the four SHALLOW metrics (Lexical, Phonetic, Morphological, Semantic) for all twelve ASR systems evaluated on the ten speech corpora, as well as their corpus-averaged values. Below, we highlight key patterns that underscore the complementary diagnostic power of SHALLOW beyond WER alone.

---

[21] https://languagetool.org/http-api/

[22] https://huggingface.co/sentence-transformers/nli-roberta-base-v2

[23] https://huggingface.co/facebook/bart-large-mnli

Table 7: WER and SHALLOW metrics evaluated on all datasets. Dataset classes are indicated as: Standard Speech , Challenging Acoustic , Heavily-Accented , Specialized Domains , and AVG (overall average) . Best results per dataset underlined, best results on average in **bold**.

| Dataset | Metrics | HuB | MMS | W-Lv2 | Canary | W-Lv3 | Parakeet | SALM. | Q2A | Granite | Kimi | Q2.5O | Phi4 |
|---|---|---|---|---|---|---|---|---|---|---|---|---|---|
| CHiME-6 | WER | 59.41 | 57.30 | 32.43 | 34.16 | 30.25 | 29.23 | 136.93 | 30.93 | 41.08 | 33.59 | 29.92 | 29.42 |
| | LF | 24.20 | 24.46 | 15.16 | 13.20 | 14.76 | 13.80 | 18.53 | 11.27 | 13.84 | 17.90 | 13.56 | 15.3 |
| | PF | 53.39 | 55.66 | 33.20 | 32.76 | 30.89 | 33.49 | 38.85 | 33.40 | 33.41 | 42.84 | 32.92 | 37.36 |
| | ME | 37.32 | 40.27 | 18.34 | 19.00 | 17.28 | 20.01 | 22.10 | 18.46 | 18.44 | 23.30 | 19.04 | 21.29 |
| | SE | 48.02 | 51.30 | 26.88 | 29.45 | 25.17 | 27.78 | 32.31 | 27.79 | 27.15 | 32.83 | 25.26 | 30.43 |
| CORAAL | WER | 45.05 | 52.74 | 22.85 | 16.58 | 19.96 | 22.47 | 75.08 | 27.34 | 22.56 | 24.16 | 22.89 | 23.67 |
| | LF | 15.82 | 19.32 | 12.94 | 7.77 | 10.20 | 9.56 | 12.31 | 7.49 | 8.79 | 12.02 | 8.31 | 10.39 |
| | PF | 40.57 | 44.55 | 28.19 | 21.24 | 24.49 | 25.59 | 29.14 | 25.73 | 25.23 | 35.22 | 27.23 | 30.64 |
| | ME | 32.35 | 37.88 | 17.11 | 14.01 | 14.68 | 17.33 | 17.54 | 16.26 | 15.22 | 19.53 | 16.24 | 18.14 |
| | SE | 36.35 | 44.30 | 23.08 | 18.28 | 19.78 | 21.12 | 23.08 | 20.07 | 20.43 | 25.69 | 20.63 | 24.15 |
| CV16-Accent | WER | 96.02 | 18.43 | 20.56 | 8.08 | 11.37 | 5.71 | 46.26 | 90.30 | 6.28 | 6.87 | 6.30 | 6.52 |
| | LF | 29.85 | 5.70 | 4.23 | 2.50 | 3.11 | 1.71 | 10.72 | 26.3 | 1.93 | 2.23 | 2.02 | 2.09 |
| | PF | 69.06 | 11.75 | 10.49 | 6.63 | 8.12 | 4.43 | 43.42 | 59.95 | 5.45 | 6.10 | 5.61 | 5.66 |
| | ME | 51.16 | 18.67 | 9.97 | 8.39 | 8.80 | 6.13 | 22.24 | 38.17 | 6.04 | 6.60 | 6.07 | 6.73 |
| | SE | 79.05 | 16.39 | 11.55 | 8.80 | 9.52 | 5.58 | 39.71 | 68.30 | 6.56 | 6.56 | 6.40 | 6.36 |
| GigaSpeech | WER | 21.13 | 22.95 | 15.52 | 13.79 | 13.71 | 11.37 | 71.62 | 11.93 | 18.85 | 12.64 | 12.35 | 12.39 |
| | LF | 10.61 | 14.58 | 13.91 | 5.31 | 13.41 | 6.37 | 12.77 | 5.26 | 7.38 | 13.28 | 7.06 | 13.02 |
| | PF | 26.31 | 34.87 | 31.44 | 16.12 | 29.16 | 16.88 | 27.47 | 16.36 | 17.55 | 31.77 | 19.65 | 30.43 |
| | ME | 19.77 | 24.71 | 16.53 | 10.15 | 15.62 | 10.55 | 15.91 | 10.09 | 10.69 | 16.61 | 11.86 | 16.31 |
| | SE | 21.42 | 27.88 | 22.95 | 13.04 | 22.28 | 12.81 | 22.31 | 12.32 | 14.59 | 23.64 | 13.67 | 21.49 |
| GLOBE-v2 | WER | 96.01 | 12.66 | 2.89 | 3.25 | 1.57 | 1.17 | 3.66 | 4.92 | 1.47 | 2.09 | 3.28 | 2.68 |
| | LF | 30.13 | 4.42 | 0.95 | 1.17 | 0.58 | 0.46 | 1.27 | 1.61 | 0.54 | 0.74 | 1.02 | 1.01 |
| | PF | 66.80 | 9.4 | 2.89 | 3.39 | 2.00 | 1.24 | 4.34 | 6.68 | 1.94 | 3.54 | 4.69 | 3.52 |
| | ME | 52.76 | 14.23 | 2.73 | 3.76 | 1.96 | 1.50 | 4.08 | 5.19 | 1.73 | 2.34 | 3.68 | 3.11 |
| | SE | 78.55 | 11.91 | 2.87 | 3.84 | 1.87 | 1.18 | 4.34 | 4.79 | 1.74 | 2.25 | 2.84 | 3.03 |
| LibriSpeech | WER | 3.51 | 7.95 | 6.15 | 3.88 | 3.98 | 2.62 | 4.94 | 3.98 | 2.98 | 2.75 | 3.46 | 3.83 |
| | LF | 1.29 | 2.88 | 2.45 | 1.48 | 1.48 | 1.00 | 1.89 | 1.42 | 1.13 | 1.16 | 1.35 | 1.56 |
| | PF | 4.48 | 8.7 | 7.87 | 5.31 | 5.24 | 3.54 | 7.46 | 5.38 | 4.46 | 4.35 | 5.22 | 5.47 |
| | ME | 5.68 | 11.44 | 7.58 | 5.92 | 5.55 | 4.35 | 7.09 | 5.71 | 4.47 | 4.42 | 5.33 | 5.80 |
| | SE | 3.51 | 8.81 | 7.19 | 5.02 | 4.63 | 2.95 | 6.60 | 4.58 | 3.61 | 3.40 | 3.96 | 4.88 |
| MyST | WER | 21.98 | 28.72 | 20.3 | 20.99 | 19.33 | 13.38 | 34.46 | 18.28 | 18.29 | 17.64 | 20.96 | 14.31 |
| | LF | 9.11 | 12.09 | 6.89 | 6.16 | 6.78 | 5.61 | 7.38 | 5.33 | 5.79 | 7.45 | 6.52 | 6.54 |
| | PF | 24.84 | 29.42 | 20.38 | 22.72 | 20.19 | 17.33 | 20.28 | 18.76 | 17.86 | 22.95 | 21.63 | 18.73 |
| | ME | 20.45 | 25.33 | 12.37 | 13.34 | 12.34 | 11.65 | 13.18 | 12.36 | 11.54 | 15.00 | 13.80 | 12.30 |
| | SE | 19.35 | 26.6 | 13.97 | 19.20 | 13.84 | 12.50 | 15.14 | 13.58 | 12.63 | 14.83 | 14.28 | 13.37 |
| SpeechOcean | WER | 37.98 | 47.04 | 25.37 | 25.35 | 21.16 | 23.90 | 25.98 | 15.66 | 24.70 | 19.92 | 13.48 | 12.88 |
| | LF | 12.98 | 15.45 | 8.19 | 8.99 | 7.46 | 8.14 | 7.04 | 5.02 | 8.41 | 6.15 | 4.43 | 4.27 |
| | PF | 21.37 | 27.75 | 16.77 | 17.24 | 16.17 | 16.76 | 15.29 | 13.64 | 16.18 | 15.21 | 9.83 | 9.32 |
| | ME | 25.30 | 32.89 | 15.28 | 17.15 | 14.69 | 16.67 | 14.88 | 12.20 | 15.3 | 14.59 | 10.95 | 10.92 |
| | SE | 31.04 | 41.11 | 22.43 | 24.92 | 21.31 | 23.69 | 20.81 | 16.86 | 22.34 | 18.37 | 14.26 | 13.76 |
| TEDLIUM | WER | 14.26 | 17.91 | 18.22 | 10.29 | 10.06 | 10.17 | 591.01 | 9.41 | 10.24 | 8.12 | 9.18 | 9.13 |
| | LF | 7.04 | 8.59 | 6.81 | 5.66 | 6.28 | 5.37 | 61.22 | 5.40 | 5.93 | 5.71 | 5.61 | 5.75 |
| | PF | 27.34 | 31.62 | 24.24 | 22.96 | 23.91 | 22.42 | 75.70 | 23.97 | 23.90 | 25.99 | 23.30 | 25.87 |
| | ME | 16.88 | 20.11 | 12.00 | 12.25 | 11.45 | 11.87 | 40.24 | 12.17 | 11.85 | 13.02 | 12.63 | 11.50 |
| | SE | 25.00 | 26.23 | 20.73 | 22.42 | 20.75 | 21.62 | 61.35 | 21.95 | 21.99 | 21.45 | 21.32 | 20.99 |
| VoxPopuli | WER | 14.05 | 8.75 | 26.92 | 6.22 | 10.57 | 5.42 | 9.21 | 7.17 | 5.65 | 7.53 | 5.77 | 5.86 |
| | LF | 4.59 | 2.77 | 9.28 | 2.02 | 3.30 | 1.75 | 2.79 | 2.15 | 1.86 | 2.54 | 1.83 | 1.90 |
| | PF | 21.48 | 15.66 | 28.37 | 12.99 | 17.29 | 11.59 | 17.04 | 14.30 | 12.05 | 16.52 | 12.45 | 12.37 |
| | ME | 13.88 | 9.88 | 19.54 | 6.55 | 8.97 | 5.80 | 8.14 | 7.06 | 6.02 | 7.57 | 5.98 | 6.09 |
| | SE | 10.69 | 6.54 | 22.09 | 4.88 | 8.22 | 4.11 | 6.64 | 5.25 | 4.60 | 5.77 | 4.47 | 4.41 |
| AVG | WER | 40.94 | 27.45 | 19.12 | 14.26 | 14.20 | 12.54 | 99.92 | 21.99 | 15.21 | 13.53 | 12.76 | **12.07** |
| | LF | 14.56 | 11.02 | 8.08 | 5.43 | 6.74 | 5.38 | 13.59 | 7.13 | 5.56 | 6.92 | **5.17** | 6.18 |
| | PF | 35.56 | 26.94 | 20.38 | 16.14 | 17.75 | **15.33** | 27.90 | 21.82 | 15.80 | 20.45 | 16.25 | 17.94 |
| | ME | 27.56 | 23.54 | 13.15 | 11.05 | 11.13 | 10.59 | 16.54 | 13.77 | **10.13** | 12.29 | 10.56 | 11.22 |
| | SE | 35.29 | 26.11 | 17.37 | 14.99 | 14.74 | 13.33 | 23.23 | 19.55 | 13.56 | 15.48 | **12.71** | 14.37 |

### D.1 MODEL-LEVEL TRADE-OFFS

**Encoder-decoder variants**  Whisper Large-v2 and Large-v3 demonstrate balanced performance across SHALLOW dimensions, with scores that avoid extreme values in any single category (PF ≈ 18–20, ME ≈ 11–13, SE ≈ 15–17). While their WER scores (19.12% and 14.20%) suggest modest differences in overall accuracy, SHALLOW metrics reveal remarkably consistent error profiles; neither model exhibits the sharp dimensional trade-offs seen in other architectures. This balanced hallucination behavior reflects their encoder-decoder design, which integrates acoustic and linguistic processing without strongly prioritizing either phonetic fidelity or semantic coherence.

**Encoder–transducer models**  Parakeet delivers the lowest phonetic fabrication score (PF = 15.33) and very competitive morphological, lexical, and semantic error rates. This highlights its architectural strength in jointly optimizing acoustic feature encoding and token prediction, enabling more precise word boundary detection and dependency modeling, which in turn minimizes both surface-level confusions and deeper structural distortions at comparable WER levels.

**Multimodal SpeechLLMs**  Phi4 and Qwen2.5Omni achieve very low average WER (12.07% and 12.76%, respectively), yet they do not uniformly minimize hallucination metrics. Phi4, for example, has higher Lexical Fabrication (6.18) and Semantic Error (14.37) than Qwen2.5Omni (LF = 5.17, SE = 12.71), revealing divergent error profiles despite similar WER. SALMONN presents a different failure pattern: despite being designed as a multimodal SpeechLLM with strong language modeling capabilities, it exhibits catastrophic WER (99.92%) while failing to leverage its architectural advantages; its hallucination scores remain comparable to simpler encoder-only models rather than aligning with the semantic coherence demonstrated by other modern SpeechLLM models. This suggests fundamental transcription failures that prevent the model from utilizing its linguistic capabilities.

### D.2 DATASET-SPECIFIC SENSITIVITIES

**Standard Speech Conditions**  On high-quality standard speech corpora, SHALLOW metrics reveal consistent patterns that WER alone cannot capture. For example, in LibriSpeech and TEDLIUM, all systems achieve low WER (3–10%, minus a few exceptions) alongside very low lexical fabrication (LF ≤ 3% for Librispeech, ≤ 8% for TEDLIUM except for SALMONN), and slightly higher semantic and morphological errors. Phonetic fabrications are instead higher, revealing that even under ideal acoustic conditions, residual phoneme-level confusions remain the primary source of errors, an effect that WER aggregates with other error types and thus obscures.

**Noisy Conversational Speech**  On CHiME-6, all models record high phonetic fabrications (PF ≈ 31–56) and moderate morphological errors (ME ≈ 18–22, except for HuBERT and MMS showing higher values), even when WER varies from 29% (Parakeet) to 137% (SALMONN). This suggests that SHALLOW isolates phonetic breakdown as the primary failure mode under acoustic overlap, a nuance lost if only WER were considered.

**Non-Native and Accented Speech**  Accented speech datasets reveal SHALLOW's diagnostic power in isolating accent-specific challenges that WER alone obscures. On CORAAL, despite WER ranging from 17% to 75%, all models exhibit consistently high PF scores (21-45), indicating that dialectal variation primarily manifests as phonetic confusions rather than lexical fabrications or semantic distortions. This pattern persists even for models achieving reasonable WER, suggesting that accent-induced errors concentrate in the phonetic dimension, a distinction completely invisible to aggregate error metrics. The consistency of elevated PF across architectures, regardless of WER performance, demonstrates how SHALLOW isolates specific failure modes that traditional evaluation conflates with general transcription quality. Such diagnostic precision enables researchers to target accent robustness improvements at the appropriate architectural level rather than pursuing generic WER gains.

**Child Speech**  MyST's spontaneous child dialogue presents unique challenges: WER rises to 13–34% across models. While lexical fabrications remain relatively low across most models (5-7%, with the exception of HuBERT at 9% and MMS at 12%), morphological (ME ≈ 12–25%), semantic

errors (SE $\approx$ 12–23%) and phonetic fabrications (PF $\approx$ 17–29%) are substantially higher. These scores reflect disfluencies and non-standard syntax in child speech, which standard acoustic and language models struggle to parse. SHALLOW thus pinpoints that errors here are not just phonetic confusions but genuine structural and meaning distortions.

### D.3 MOTIVATION FOR SHALLOW METRICS

The patterns above demonstrate that:

1. *WER is insufficiently granular*: Models with near-identical WER can have markedly different hallucination profiles (e.g., Phi4 vs. Qwen2.5Omni).

2. *Error modes diverge by dataset*: Noisy or dialectal corpora elevate specific hallucination types (e.g., phonetic in CHiME-6, lexical in CORAAL) that WER alone cannot disentangle.

3. *Architectural trade-offs become visible*: Encoder- and decoder-centric designs show complementary strengths (acoustic vs. linguistic), which SHALLOW quantifies directly.

4. *Semantic hallucinations persist despite low WER*: SHALLOW reveals that meaningful content distortions, especially polarity flips or misattributions, can occur even when overall transcription accuracy appears high (i.e., WER is low).

These observations underscore SHALLOW's role as a *multi-dimensional* diagnostic toolkit: by decomposing ASR errors into lexical, phonetic, morphological, and semantic axes, it surfaces nuanced failure modes and informs targeted model improvement strategies that aggregate WER cannot provide.

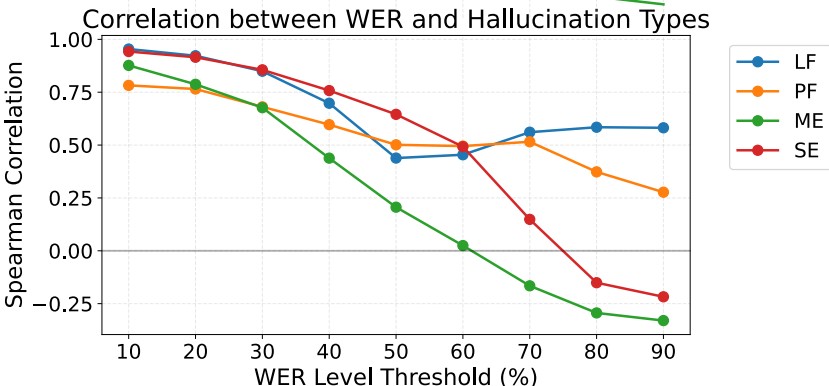

Figure 6: Spearman correlation between WER and each SHALLOW metric, computed on samples filtered by varying WER threshold levels.

## E ADDITIONAL ANALYSIS

### E.1 CORRELATION ACROSS WER THRESHOLDS

Figure 6 presents a threshold-based correlation analysis between WER and the four SHALLOW hallucination metrics: Lexical Fabrication (LF), Phonetic Fabrication (PF), Morphological Errors (ME), and Semantic Errors (SE). We compute Spearman correlation coefficients between WER and each hallucination type, restricting the analysis to model–dataset pairs with WER below increasing thresholds from 10% to 90%.

**Correlation trends.** At low WER levels (below 30–40%), all hallucination metrics are strongly correlated with WER (Spearman $\rho \geq 0.70$), indicating that when models perform well, WER changes largely reflect proportionate reductions in lexical, phonetic, and semantic errors. However, as WER increases, correlations diverge:

- LF remains moderately correlated with WER ($\rho \approx 0.60$) even at high WER, confirming its central role in contributing to raw word errors.
- PF correlation gradually decreases, indicating that phonetic hallucinations become less predictive of WER in degraded conditions.
- ME and SE exhibit sharp correlation drop-offs, eventually turning near-zero or negative (ME: $\rho < 0$ past 60% WER), showing that morphological and semantic distortion no longer track with raw WER.

These results empirically validate our core claim of the SHALLOW framework: as model performance deteriorates, WER ceases to reliably reflect specific error types, especially those involving meaning and structure, while SHALLOW retains discriminative power.

### E.2 WER-HALLUCINATION CORRELATION HEATMAP

Figure 7 displays a Spearman correlation heatmap between WER and each SHALLOW hallucination type with increasing WER thresholds (from 10% to 90%). This visualization complements the trend plot in Figure 6, offering the same underlying information but at a finer-grained, value-specific level. While the line plot emphasizes overall trends in correlation strength, the heatmap makes it easier to inspect exact correlation values across conditions.

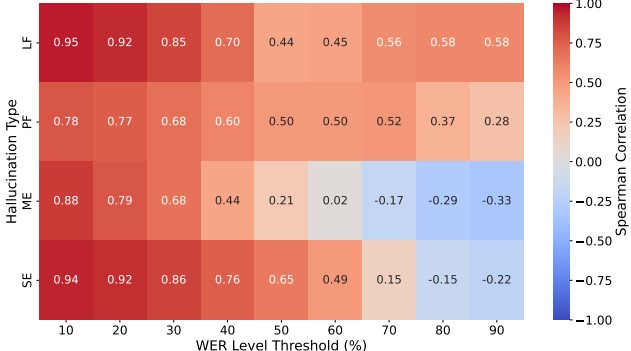

Figure 7: Spearman correlations between WER and each SHALLOW metric, computed over model–dataset pairs with WER below increasing thresholds.

**Strong alignment in low-WER regimes.** At low WER thresholds (10%-30%), all hallucination metrics exhibit strong positive correlations with WER (Spearman $\rho \geq 0.68$), with LF and SE reaching values above $0.90$ for lower WER thresholds. This indicates that under high-quality recognition conditions, changes in WER closely reflect changes across all hallucination dimensions, confirming that WER remains a reasonable proxy for error severity when models operate in near-correct regimes.

**Semantic and morphological divergence in higher-WER settings.** As WER thresholds increase beyond $40\%$, correlations with SE and ME degrade sharply. By $70\%$ WER, the correlation between WER and ME becomes negative ($\rho = -0.17$), and continues decreasing to $-0.33$ at $90\%$, indicating that morphological hallucinations become statistically decoupled, and even inversely associated, with WER under severe degradation. Semantic error correlation similarly flips sign beyond $70\%$, highlighting that meaningful distortions are no longer well-aligned with raw error rate as models deteriorate.

**Lexical and phonetic metrics remain moderately aligned.** In contrast, LF maintains a relatively stable correlation with WER (remaining above $\rho = 0.44$), even at high thresholds. This confirms that lexical fabrication contributes consistently to word-level mismatch across performance ranges. PF exhibits a gradual drop in correlation, settling at $\rho = 0.28$ at the $90\%$ WER threshold, showing a moderate but diminishing relationship.

### E.3 EXAMPLES

Table 8 shows representative reference-hypothesis pairs from each dataset for six models (Whisper Large-v3, MMS, Parakeet, SALMONN, Qwen2Audio, and Phi-4). These exemplify how WER alone can mask important differences in error types, while SHALLOW metrics reveal the specific nature of hallucinations.

## F    COMPUTATIONAL RESOURCES

All SHALLOW experiments and metric evaluations were conducted using a single NVIDIA A100 80GB GPU. This setup was sufficient for both inference over the evaluated ASR systems and full-scale metric computation across all datasets. The complete SHALLOW framework is implemented in a modular and GPU-accelerated fashion where applicable.

**Metric-wise computational complexity.**    While WER remains the most widely used metric in ASR evaluation, its simplicity comes with limited diagnostic resolution. SHALLOW metrics provide a richer decomposition of hallucination phenomena, but at the cost of increased computational overhead. Below, we outline the time complexity characteristics per metric:

- *Lexical Fabrication (LF):* Computed using insertions, deletions, and substitutions derived from Levenshtein alignment. This shares the exact operational backbone with WER and thus incurs negligible additional cost over WER.
- *Phonetic Fabrication (PF):* Based on phonetic similarity via metaphone, PF is computed per sentence pair and is computationally lightweight, with runtime on par with LF and WER.
- *Morphological Error (ME):* Involves parsing both hypothesis and reference into dependency graphs using standard syntactic parsers. This step introduces a higher per-sample cost, particularly sensitive to sentence length and syntactic complexity. Runtime grows linearly with the number of tokens and the branching factor of the parse tree.
- *Semantic Error (SE):* Relies on the computation of sentence-level embeddings (both local and global views), using lightweight transformer-based models. While embedding inference is efficient on modern hardware, SE still incurs a higher cost due to multiple similarity computations (distance and coherence).

**Edge-case robustness.**    To prevent unnecessary computation and ensure robustness, SHALLOW incorporates deterministic backoff mechanisms for degenerate cases. If either the reference or hypothesis is empty, or if the pair is exactly equal, metrics are short-circuited to return default values (e.g., zeros or maximum similarity), avoiding meaningless downstream computation.

**Runtime variability.**    End-to-end metric computation time varies as a function of (i) Number of samples in the dataset; (ii) Number of edge cases encountered; (iii) Average sentence length per hypothesis–reference pair; (iv) Linguistic complexity (which affects parsing and embedding models); and (v) Number of parallel threads that can be employed. For example, the complete evaluation of the LibriSpeech corpus ($3K$ samples) takes approximately 90 minutes on a single GPU. In contrast, larger and more heterogeneous datasets such as GigaSpeech require more time, depending on batch processing and parser throughput.

**ASR model inference.**    Inference for the evaluated ASR systems was conducted using publicly available checkpoints and libraries, all run locally on the same A100 GPU. Models with encoder-only, encoder-decoder, or encoder–transducer architectures (e.g., MMS, Whisper, and Parakeet) exhibit efficient inference times (throughput RTFx[24] $\geq$ 2300 for Parakeet), while decoder-only or instruction-tuned SpeechLLMs (e.g., Phi4, SALMONN) show longer inference latencies due to autoregressive decoding (up to 4–5$\times$ slower).

**Scalability and batching.**    SHALLOW is designed to process utterances in parallel batches where possible (e.g., embedding-based SE metrics, WER alignment). Parsing-based operations (e.g., ME) remain inherently sequential due to parser design, but can still be parallelized with thread-level concurrency.

SHALLOW incurs modest overhead over traditional WER-based pipelines, especially for metrics requiring linguistic or semantic modeling. Nonetheless, the added interpretability and diagnostic precision justify this cost, especially for applications in critical domains where error type matters more than raw accuracy. Our framework balances efficiency and detail, scaling effectively from small synthetic stress tests to full-scale benchmarks across real-world corpora.

---

[24]Throughput is measured using the RTFx metric, defined as the number of seconds of audio inferred divided by the compute time in seconds. It is the inverse of the RTF (Real Time Factor) metric.

Table 8: Examples of evaluated datasets with WER and all SHALLOW metrics.

| DS | Model | Hypothesis | Reference | WER | LF | PF | ME | SE |
|---|---|---|---|---|---|---|---|---|
| **ChiME-6** | Wv3 | o my god | thank you | 1.00 | 0.27 | 0.71 | 0.40 | 0.67 |
| | MMS | | a my gad | 0.67 | 0.20 | 0.45 | 0.40 | 0.37 |
| | Par | | o | 0.67 | 0.13 | 1.00 | 0.27 | 0.48 |
| | SALM | | i am sorry i did not catch that could you repeat it | 4.0 | 0.68 | 0.77 | 0.40 | 0.62 |
| | Q2A | | o my gosh | 0.33 | 0.10 | 0.20 | 0.32 | 0.18 |
| | Phi4 | | my god | 0.33 | 0.07 | 0.00 | 0.13 | 0.27 |
| **CORAAL** | Wv3 | jeremiah he turnt up too | jeremiah you turn to us | 0.80 | 0.24 | 0.30 | 0.45 | 0.61 |
| | MMS | | grma ict 0 | 1.00 | 0.26 | 0.60 | 0.56 | 0.81 |
| | Par | | jeremiah return to | 0.80 | 0.20 | 0.40 | 0.48 | 0.65 |
| | SALM | | jeremiah we turn to | 0.80 | 0.22 | 0.30 | 0.46 | 0.49 |
| | Q2A | | jeremy yu chang also | 1.00 | 0.28 | 0.51 | 0.58 | 0.37 |
| | Phi4 | | jeremiah you turned | 0.80 | 0.20 | 0.30 | 0.48 | 0.35 |
| **CV16-Accent** | Wv3 | queuing is something the british excel at | kiwi means something that bridges excel ads | 0.71 | 0.21 | 0.35 | 0.37 | 0.71 |
| | MMS | | kiwi knew something that bridgis excel at | 0.57 | 0.17 | 0.30 | 0.37 | 0.54 |
| | Par | | queuing is something the british excel at | 0.00 | 0.00 | 0.00 | 0.00 | 0.00 |
| | SALM | | kiwi needs something that bridges excel at | 0.57 | 0.17 | 0.34 | 0.33 | 0.54 |
| | Q2A | | kids are talking by the door | 1.00 | 0.31 | 0.58 | 0.40 | 0.83 |
| | Phi4 | | kiwis need something the british excel at | 0.29 | 0.09 | 0.12 | 0.27 | 0.35 |
| **GigaSpeech** | Wv3 | in its hold | and it is old | 1.33 | 0.43 | 0.49 | 0.40 | 0.55 |
| | MMS | | in it old | 0.67 | 0.20 | 0.25 | 0.40 | 0.53 |
| | Par | | in its hold | 0.00 | 0.00 | 0.00 | 0.00 | 0.00 |
| | SALM | | in its hole | 0.33 | 0.10 | 0.07 | 0.32 | 0.66 |
| | Q2A | | in its hole | 0.33 | 0.10 | 0.07 | 0.32 | 0.66 |
| | Phi4 | | ill it hold | 0.67 | 0.20 | 0.30 | 0.40 | 0.47 |
| **GLOBE-v2** | Wv3 | then what does she want with you | yeah i do what does she want to see | 0.71 | 0.24 | 0.50 | 0.36 | 0.53 |
| | MMS | | le azil ortas ci wol amfkesi | 1.00 | 0.29 | 0.69 | 0.64 | 0.80 |
| | Par | | nadel what does she want with you | 0.14 | 0.04 | 0.44 | 0.21 | 0.14 |
| | SALM | | that is all she wants monsieur | 0.86 | 0.24 | 0.48 | 0.40 | 0.65 |
| | Q2A | | what does she want pete | 0.43 | 0.10 | 0.53 | 0.25 | 0.45 |
| | Phi4 | | yeah the other shivaam feature | 1.00 | 0.27 | 0.76 | 0.47 | 0.83 |
| **LibriSpeech** | WV3 | she continued father fauvent | she continued for the fervent . | 1.00 | 0.32 | 0.27 | 0.30 | 0.27 |
| | MMS | | she continued father | 0.25 | 0.05 | 0.23 | 0.33 | 0.20 |
| | Par | | she continued father fauven | 0.25 | 0.08 | 0.05 | 0.33 | 0.09 |
| | SALM | | she continued further prevent | 0.50 | 0.15 | 0.24 | 0.27 | 0.25 |
| | Q2A | | she continued father frovent | 0.25 | 0.08 | 0.10 | 0.33 | 0.10 |
| | Phi4 | | she continued father prevent | 0.25 | 0.08 | 0.15 | 0.27 | 0.15 |
| **MyST** | Wv3 | because we are because we are learning about | because we have been because learning about learning things . | 0.75 | 0.25 | 0.48 | 0.38 | 0.25 |
| | MMS | | because we have been becas arling about loingthings i aaar | 1.00 | 0.33 | 0.47 | 0.50 | 0.51 |
| | Par | | because we have been because running about living things but that | 1.00 | 0.32 | 0.53 | 0.40 | 0.63 |
| | SALM | | because we have been because we have been because we have been because we have been [...] | 24.5 | 0.63 | 0.82 | 0.40 | 0.34 |
| | Q2A | | because we have to because learning about doing things but the | 1.00 | 0.32 | 0.48 | 0.38 | 0.37 |
| | Phi4 | | because we have been because learning about living things but but but | 1.13 | 0.35 | 0.56 | 0.38 | 0.37 |
| **SpeechOcean** | wv3 | alice give up boxing | and skip that book scene | 1.25 | 0.40 | 0.60 | 0.40 | 0.73 |
| | MMS | | aris gave tha buksin | 1.00 | 0.30 | 0.39 | 0.58 | 0.68 |
| | Par | | alex gave up boxing | 0.50 | 0.15 | 0.38 | 0.46 | 0.51 |
| | SALM | | aris give up boxing | 0.25 | 0.08 | 0.06 | 0.22 | 0.14 |
| | Q2A | | let us give up boxing | 0.50 | 0.18 | 0.47 | 0.29 | 0.25 |
| | Phi4 | | alice gave up boxing | 0.25 | 0.07 | 0.05 | 0.46 | 0.09 |
| **TEDLIUM** | Wv3 | and i can twist that around i am sorry if you are getting queasy look away do not look at the thing | and i can twist that around i am sorry if you are getting queasy look away do not look at the thing | 0.00 | 0.00 | 0.00 | 0.00 | 0.00 |
| | MMS | | and i can twist that around i am sorry if you are getting queezy look awaydo not look at thei | 0.23 | 0.06 | 0.12 | 0.19 | 0.08 |
| | Par | | and i can twist that around i am sorry i do not if you are getting queasy look away do not look at the thing | 0.14 | 0.06 | 0.25 | 0.12 | 0.22 |
| | SALM | | thank you for tuning in to our radio show today we are going to be discussing the effects of marijuana on the brain [...] | 5.64 | 0.66 | 0.76 | 0.41 | 0.59 |
| | Q2A | | and i can twist that around i am sorry i if you are getting queezy look away do not lok at te thing | 0.18 | 0.06 | 0.20 | 0.27 | 0.09 |
| | Phi4 | | and i can twist that around i am sorry if you are getting queasy look away do not look at the | 0.05 | 0.01 | 0.04 | 0.11 | 0.04 |
| **VoxPopuli** | Wv3 | i appreciate very much what you said but can you make sure that once you foresee this kind of simulation today that you invite some of the people who were actually in mumbai because it could give you some insight | okay | 1.00 | 0.20 | 0.83 | 0.40 | 0.84 |
| | MMS | | ie very much what you said but can you make sure once you foresee this kind of simulation todays that you invite some of the people which were actually in mumbay i think it could be given you some insid | 0.28 | 0.09 | 0.38 | 0.28 | 0.13 |
| | Par | | i appreciate very much what you said but can you make sure once you foresee this kind of simulation 2 days that you invite some of the people which were actually in mumbai i think it could give you some insight | 0.15 | 0.05 | 0.21 | 0.16 | 0.24 |
| | SALM | | appreciate very much what you said but can you make sure once you foresee this kind of simulation to days that you invite some of the people which were actually in mumbai i think it could give us some insight | 0.20 | 0.07 | 0.38 | 0.15 | 0.16 |
| | Q2A | | i appreciate very much what you said but can you make sure once you foresee this kind of simulation to days that you invite some of the people who were actually in mumbai i think it could give you some insight | 0.13 | 0.04 | 0.28 | 0.12 | 0.19 |
| | Phi4 | | but can you make sure once you foresee this kind of simulation today that you invite some of the people which were actually in mumbai i think it could give you some insight | 0.28 | 0.07 | 0.45 | 0.20 | 0.11 |

