# OpenReview forum: "Hallucination Benchmark for Speech Foundation Models"
_ICLR.cc/2026/Conference — Submitted to ICLR 2026_

### Official Review · Reviewer_vMKW · 2025-10-28

**Soundness:** 3
**Presentation:** 3
**Contribution:** 3
**Rating:** 6
**Confidence:** 3

**Summary:**

This paper identifies a lack of standardized methods for systematically categorizing and measuring hallucinations in existing ASR research. To address this gap, it introduces SHALLOW, a benchmark framework that categorizes ASR errors into four complementary dimensions. Through systematic analysis, the authors demonstrate that the proposed four measurements are distinct and mutually complementary. Furthermore, evaluations based on the proposed metric provide a detailed analysis of the critical weaknesses in existing ASR models. Given that previous ASR hallucination studies primarily relied on WER as a key metric, this research offers significant value to the field.

**Strengths:**

- The study effectively identifies and critiques the reliance on aggregate metrics like WER in current evaluation practices, highlighting significant issues in existing ASR hallucination research. The motivation for the experiments is well-structured and the authors address this problem with precision and timeliness.
- The authors propose the SHALLOW benchmark framework, categorizing and quantifying hallucination phenomena in ASR across four complementary axes: lexical, phonetic, morphological, and semantic. Through several experiments, the authors demonstrated distinct and complementary nature of these axes, validating the proposed classification system.
- The study conducted robust experiments across a diverse range of models and datasets. The four proposed axes enable a more detailed analysis of ASR models, previously reliant only on WER, allowing for a comprehensive examination of their strengths and weaknesses. The study reveals hallucinations undetectable by WER alone and empirically demonstrates the ability of SHALLOW to uncover them.

**Weaknesses:**

- Some necessary details appear to be omitted in the design of the metric.
    - The weights in most of the proposed metrics seem to be chosen arbitrarily. Although the authors claim to describe these details in Appendix B, I can hardly find such details. For the proposed metric to serve as a standard for measuring hallucination, I believe the rationale for the coefficients in Section 3 must be clarified.
    - This issue appears in all four proposed metrics, and clearer justification would be needed.
        - e.g., Why choose 0.5 for ( r_i ), 0.3 for ( r_s ), and 0.2 for ( r_d ) in Equation 1? How were these weights determined?
        - e.g., Why use a naive average for calculating PF in Equation 2, while applying weights of 0.4 for SD and 0.6 for GE in Equation 4?
    - (Section 3.3) How are the errors in grammar (E_Gr), spelling (E_Sp), and punctuation (E_Pu) measured?
    - (Section 3.3) Which dependency parser was used to measure morphological errors?

- (Sections 3.3-3.4) When using neural models as metrics, it is crucial to justify their reliability.
    - For instance, the reliability of an NLI model's accuracy on out-of-distribution test sets must be addressed. The study should explore how this can be trusted.
    - The reliability of methods like BERTScore also requires comprehensive validation. The authors mentioned using BERT-based models in line 282, but the specific models utilized are not specified. The authors should clarify which models were used and how their reliability was validated.
    - Moreover, encoder-based models typically assess sentence-level similarity, yet SHALLOW apply them to measure unigram, bigram, and trigram alignment. While not inherently incorrect, this application demands performance validation.
    - Without such analysis, the measured semantic error might not constitute a meaningful measure.

**Questions:**

- How do the authors define the SHALLOW benchmark framework?
    - Does the dataset covered by SHALLOW refer to the datasets experimented on in Section 4, or the verification dataset constructed in Section 1?
    - Without a clearly defined dataset range, SHALLOW might be more appropriately termed a metric rather than a benchmark.
- Upon examining each metric, the low correlation between WER and the four metrics is well-acknowledged. However, Table 2 shows that models with generally low WER also tend to exhibit low hallucination tendencies. Could WER, therefore, be seen as a comprehensive metric for measuring hallucination? What are the authors' thoughts on this?
- Why does the SALMONN language model show a WER of 99.92 in Table 2? This appears to differ significantly from results reported in previous studies. Can the authors provide examples of SALMONN's generated outputs and justify such results?
- Are the performances reported in Table 2 averaged over all datasets evaluated in Table 4? Did the authors use a combined dataset for evaluation?
- Regarding phonetic fabrication, is the Philips (2000) method for metaphone transformations sufficiently reliable? I acknowledge that several neural models have been proposed for metaphone transformations, surpassing traditional algorithms. How do the authors address potential transcription errors when using this algorithm-based approach and metric?

---

### Official Review · Reviewer_iPx2 · 2025-10-30

**Soundness:** 4
**Presentation:** 3
**Contribution:** 4
**Rating:** 8
**Confidence:** 4

**Summary:**

This paper introduces a new benchmark for ASR evaluation by accounting for lexical, phonetic, morphological and semantic hallucinations. Along these four directions, the authors mathematically formulate quantifications which is called the SHALLOW benchmark. A synthetic dataset is also curated keeping in mind the above four types of errors which further facilitates fine-grained evaluation of ASR systems. Comprehensive evaluations of various ASR models on various datasets using the SHALLOW metrics show the effectiveness of the proposed approach.

**Strengths:**

1. The problem tackled is an important one.
2. The authors have carefully considered how to best evaluate the ASR systems and come up with four dimensions in their benchmark: lexical, phonetic, morphological and semantic.
3. Novel evaluation metrics are introduced.
4. Comprehensive evaluations done on different datasets and ASR models.

**Weaknesses:**

1. A lot of important details (like the methodology for the creation of the synthetic dataset) has been delegated to the appendix.
2. The paper sometimes has clarity issues which should be fixable (see next section).

**Questions:**

1. The authors mention the creation of the synthetic dataset. It would be good to have some quantitative evaluations of different ASR models on this dataset in the main text.
2. What is meant by "metric vectors" mentioned in line 139-140? Please provide more details in the main text on how these are computed to plot the t-SNE.
3. In equation (1), the weights on $r_i, r_s$ and $r_d$  should sum to $3.0$ instead of $1.0$ as an analogy to $WER = \frac{N_i + N_s + N_d}{N} = r_i + r_s + r_d$.
4. Please provide more details about Hamming distance, Levenshtein distance and Jaro-Winkler similarity in the main text.
5. How do you automate the computation of $E_{Gr}$, $E_{Sp}$ and $E_{Pu}$ while computing the Morphological errors.
6. Please explain the difference between semantic distance and semantic coherence.
7. How are the weights chosen for aggregated semantic error score.

---

### Official Review · Reviewer_NAG3 · 2025-11-01

**Soundness:** 1
**Presentation:** 2
**Contribution:** 2
**Rating:** 2
**Confidence:** 4

**Summary:**

This paper introduces SHALLOW, a set of new evaluation metrics for speech recognition beyond word error rate (WER) with a goal of measuring hallucination. The authors define hallucination to be "fluent and coherent transcriptions produced by neural ASR models that are completely unrelated to the underlying acoustic input" (line 11-13). They propose to measure hallucination in 4 dimensions: lexical fabrication, phonetic fabrication, morphological errors, semantic errors. By comparing 12 neural ASR models across several datasets, the authors found that SHALLOW correlate closely when WER is low, but the correlation is much lower when the WER is high.

**Strengths:**

- Originality: This paper proposed a novel taxonomy of hallucination by ASR models.
- Quality: This paper evaluated the proposed metrics on a wide collection of ASR models and test sets in order to demonstrate its characteristics.
- Clarity: Most parts of this paper is well organized and clearly written.
- Significance: As WER decreases, it is ever more important to get a more refined understanding of the sources of ASR errors. This paper is an attempt in that direction.

**Weaknesses:**

-   The proposed metrics, while intuitively sensible, are not backed with empirical evidence for their effectiveness in measuring the claimed properties.
    -   Good metrics (e.g. BLEU, BLEURT, WER) all follow the same recipe:
        1.  First a grounded cost objective is established: BLEU and WER aim to approximate the cost of human post editing of machine generated text; BLEURT aims to approximate the human judgement of translation quality.
        2.  Hypotheses with different costs are collected, and scored using the proposed metrics.
        3.  Correlation between the cost and the proposed metric is used as evidence of the effectiveness of the proposed metric.
    -   In this paper, while there are reasonably well defined error categories, none of the proposed metrics has been evaluated to check whether they actually accurately measure these categories of errors.
-   The overall utility of the proposed metrics are not well demonstrated. This paper reports that when WER is low, SHALLOW closely correlates with WER. Only when the WER is high, the correlation decreases. However, SHALLOW makes heavy use of pretrained NLP models, many of which may not be reliable for high WER inputs (due to these being far from typical text seen during training). Thus SHALLOW numbers for high WER inputs might not be as reliable overall as those for low WER inputs. However, in the low WER case, SHALLOW does not appear to differentiate different models well enough.

**Questions:**

I do not have further questions.

---

### Official Review · Reviewer_fExH · 2025-11-03

**Soundness:** 3
**Presentation:** 3
**Contribution:** 2
**Rating:** 2
**Confidence:** 4

**Summary:**

This paper introduces SHALLOW (Speech HALLucination OvervieW), a benchmark framework for evaluating hallucinations in ASR systems across four dimensions: lexical fabrications, phonetic fabrications, morphological errors, and semantic errors. The authors evaluate 12 ASR models across 10 diverse speech datasets and demonstrate that SHALLOW metrics reveal error patterns that WER alone cannot distinguish, particularly under degraded acoustic conditions. A synthetic validation dataset of 1,050 pairs confirms metric specificity. Key findings show that SHALLOW metrics correlate with WER when recognition quality is high but decouple significantly as WER increases, with correlations dropping from >0.8 to negative values at high error rates.

**Strengths:**

1. **Comprehensive multi-dimensional framework**: Decomposing errors into lexical, phonetic, morphological, and semantic dimensions provides interpretable diagnostic capability that goes beyond binary hallucination detection (Frieske & Shi) and complements HER-based approaches (Atwany et al.)

2. **Validation**: The 1,050-pair controlled dataset (Table 1, Figure 4) with isolated error types demonstrates metric specificity through t-SNE separability, addressing a gap in prior work that relied on either heuristics or human judgment alone.

3. **Correlation breakdown insight**: Figure 3's demonstration that SHALLOW metrics decouple from WER as error rates increase (correlations dropping from >0.8 to <0 at high WER) is a valuable empirical contribution showing when traditional metrics fail

4. **Extensive empirical coverage**: Evaluation across 12 models and 10 datasets spanning standard, noisy, accented, and specialized domains provides robust evidence across diverse conditions

5. **Deterministic and reproducible**: Unlike LLM-based approaches (Atwany et al.), SHALLOW's rule-based metrics enable exact reproduction without API costs or model drift concerns

6. **Medical case study impact**: Table 3 concretely illustrates high-stakes failures where low WER masks critical semantic inversions ("can not" → "can"), demonstrating real-world relevance beyond academic benchmarks

**Weaknesses:**

1. **Lack of technical novelty**: No novel algorithms, architectures, or theoretical frameworks are introduced. The contribution is primarily engineering—packaging existing tools into a unified benchmark—rather than methodological innovation. This contrasts with Atwany et al., who introduce HER as a new metric with distribution shift theory, and Frieske & Shi, who propose perturbation-based hallucination detection methods.

2. **Insufficient positioning against concurrent work**: The paper cites Atwany et al. (2025) and Frieske & Shi (2024) but provides no empirical comparison. Critical missing experiments:
- Apply SHALLOW, HER (Atwany), and heuristic baseline (Frieskeet et al.) to the same test set
- Compare computational cost: SHALLOW's tool pipeline vs. LLM API calls
- Validate whether SHALLOW's dimensions correlate with Atwany's coarse/fine-grained categories
- Show which approach better predicts human judgments of hallucination severity

3. **Weighting schemes lack principled justification**: Weights (e.g., 0.5/0.3/0.2 for LF, 0.4/0.6 for ME) appear arbitrary. The claim of validation "through analysis of error patterns" (Appendix B) is descriptive rather than optimized. Unlike Atwany's distribution shift correlation (α=0.91) or Frieske's empirically tuned thresholds, SHALLOW provides no sensitivity analysis or optimization procedure

4. **English-only severe limitation**: All evaluation is English-only despite claims about "ASR hallucinations" generally. Semantic metrics explicitly require "language-specific NLP models," yet morphological errors are noted as important for "low-resource languages where morphological richness carries semantic distinctions" (none evaluated). This is a critical gap that Atwany's work acknowledges but doesn't address either

5. **Tool dependency uncharacterized**: Reliance on LanguageTool, spaCy, Berkeley parser, BERT/RoBERTa introduces failure modes not explored. When do parsers fail on disfluent speech? How do embeddings handle domain jargon? Atwany's LLM-based approach has similar dependencies but provides verbalized confidence scores (Tables 2-3) to quantify uncertainty

6. **No architectural insights**: Unlike Atwany et al., who demonstrate that encoder-only models exhibit lower HER/WER ratios and that model size affects HER non-monotonically, SHALLOW provides purely diagnostic results without explaining *why* certain architectures produce certain hallucination patterns

7. **Synthetic validation may not transfer**: The clean t-SNE separation (Figure 2) on synthetic data may not reflect real ASR errors where hallucination types are entangled. Frieske & Shi found that dataset noise (unique-unique, repeat-repeat) creates specific error distributions does SHALLOW capture these patterns?

8. **Unclear actionability**: Practitioners receive four numbers (LF/PF/ME/SE) with no interpretation guidance. When is PF acceptable vs. problematic? How should one trade off dimensions? Atwany's HER provides a single interpretable metric; Frieske's binary classification is simpler. SHALLOW's richness may hinder adoption without decision frameworks

**Questions:**

1. Can the authors compare SHALLOW with Atwany et al.'s HER and Frieske et al.'s heuristic baseline on a shared test set (e.g., LibriSpeech, CHiME-6)? How do the approaches correlate? Which best predicts human hallucination judgments?

2. Tables 4-6 show raw components—can authors report SHALLOW scores under alternative weighting schemes (equal weights, domain-optimized)? Was any optimization performed, or are weights purely intuitive?

3. Do SHALLOW's four dimensions map to Atwany's fine-grained categories (Hallucination Error, Phonetic Error, Language Error, Oscillation Error, No Error)? Can authors show correspondence on overlapping examples?

4. When do spaCy parsing or BERT embeddings fail? Can you characterize error rates for the measurement infrastructure itself? What happens on heavily disfluent speech (e.g., CORAAL, MyST)?

5. Does the t-SNE separability (Figure 2) hold on real ASR outputs? Frieske et al. showed that training data noise (UU, RR) affects error types. Can SHALLOW discriminate these patterns?

7. What's SHALLOW's total cost (compute + tools) and comparison with other works in terms of cost and efficiency?

8. Frieske et al. use perturbations (noise injection) to detect hallucinatory models at test time. Does SHALLOW enable similar detection by measuring dimension changes under perturbation?

10. Given a SHALLOW profile (high PF, low LF), what model improvements would you recommend? Can the authors provide heuristics or decision trees?

---

### Meta-Review · Area_Chair_ZUzd · 2026-01-07

**Summary:**

This paper introduces a benchmark called SHALLOW for evaluating lexical, phonetic, morphological, and semantic hallucinations in automatic speech recognition (ASR) systems. The authors present a synthetically generated validation set to support this evaluation framework. The paper includes experiments that investigate the correlation between word error rate (WER) and various types of hallucination errors across 12 different ASR models, which were trained on many different speech datasets.

**Reviewer Concerns:**

**Addressed**

None of the reviewer concerns were addressed by the authors.

**Unaddressed**

1. The paper lacks empirical comparison with related benchmarks in the field, making it difficult to assess the relative value and performance of SHALLOW compared to existing evaluation frameworks.
2. The intuitions and foundations behind the proposed hallucination metrics themselves remain unclear. While the metrics are automatic metrics, they do not appear to follow the rigorous standards established by previous automatic evaluation metrics, such as BLEURT.
3. The benchmark is limited to English only, which restricts its applicability and impact for the broader multilingual ASR research.
4. The paper is missing important methodological and implementation details, which make the work difficult to follow and reproduce.

**Reviewer Scores:**

There was no rebuttal so no way for reviewers to alter their scores.

---

### Decision · Program_Chairs · 2026-01-26

Reject